# Review article: Inferring permafrost and permafrost thaw in the mountains of the Hindu Kush Himalaya region

Stephan Gruber[1], Renate Fleiner[2], Emilie Guegan[3], Prajjwal Panday[4], Marc-Olivier Schmid[2], Dorothea Stumm[2], Philippus Wester[2], Yinsheng Zhang[5], Lin Zhao[6]

[1]Carleton University, Ottawa, Canada
[2]International Centre for Integrated Mountain Development (ICIMOD), Kathmandu, Nepal
[3]Norwegian University of Science and Technology, Trondheim, Norway
[4]Nichols College, Dudley, MA, USA
[5]Institute of Tibetan Plateau Research, Chinese Academy of Sciences, Beijing, China
[6]Cold and Arid Regions Environmental and Engineering Research Institute, Chinese Academy of Sciences, Lanzhou, Gansu, China

*Correspondence to*: Stephan Gruber (stephan.gruber@carleton.ca)

**Abstract.** The cryosphere reacts sensitively to climate change, as evidenced by the widespread retreat of mountain glaciers. Subsurface ice contained in permafrost is similarly affected by climate change, causing persistent impacts on natural and

human systems. In contrast to glaciers, permafrost is not observable spatially and therefore its presence and possible changes are frequently overlooked. Correspondingly, little is known about permafrost in the mountains of the Hindu Kush Himalaya region, despite permafrost area exceeding that of glaciers in nearly all countries. Based on evidence and insight gained mostly in other permafrost areas globally, this review provides a synopsis on what is known or can be inferred about permafrost in the mountains of the HKH region. Given the extreme nature of the environment concerned, it is to be expected

that the diversity of conditions and phenomena encountered in permafrost exceed what has previously been described and investigated. We further argue that climate change in concert with increasing development will bring about diverse permafrost-related impacts on vegetation, water quality, geohazards, and livelihoods. To better anticipate and mitigate these effects, a deepened understanding of high-elevation permafrost in subtropical latitudes as well as the pathways interconnecting environmental changes and human livelihoods are needed.

**Introduction**

The mountain ranges spanning Afghanistan, Bhutan, China, India, Myanmar, Nepal and Pakistan, here referred to as the Hindu Kush Himalaya (HKH) region, are the source of the ten largest Asian river systems and provide water, ecosystem services and the basis for livelihoods to an estimated population of 200 million people. This region of more than four million km$^2$ is mostly comprised of high-elevation rugged terrain between 25º and 40º northern latitude (Figure 1 AB). It includes

the Qinghai-Tibetan plateau (QTP), the Hindu-Kush, Himalaya, and parts of the Pamir and Karakoram mountain ranges. The HKH region has a glacier cover of around 70 thousand km$^2$. While glaciers, glacier changes, and the impacts of those

changes are comparably well investigated (Bolch et al., 2012; Kääb et al., 2012; Yao et al., 2012), permafrost (commonly defined as soil or rock remaining below 0ºC for more than two consecutive years) and the potential impacts of its thaw remain largely unknown for most of the HKH.

We argue that knowledge about permafrost in the HKH is important, because of its large areal abundance (Figure 1 E), large population, and likely thaw-related impacts in the near future that can be inferred from other areas globally. Possible impacts include changed frequency and unexpected location of landslides, as well as changes to vegetation, runoff patterns, and water quality. Cumulatively and in a global context, these impacts will likely be less severe than those caused by e.g. earthquakes, drought or tropical storms. They will, however, be widespread, persistent and often unexpected, especially for populations on or near high-elevation terrain, whose livelihoods are strongly dependent on the surrounding land and ecosystems. While permafrost research and engineering around linear infrastructure on the QTP is developed and globally visible (Yang et al., 2010a; Zhang et al., 2008), there is virtually no local research capacity and very few measurements concerning permafrost in other areas of the HKH, more pronounced even than a similar lack of measurements identified with respect to glaciers and meteorology (Bolch et al., 2012).

With this review, we distil the current state of knowledge on the distribution and characteristics of permafrost in the HKH as well as on possible consequences of its thaw. In this, we rely strongly on research from other regions to infer conditions in the HKH because little local information on permafrost is available. This review comprises explanation and referencing to make it accessibility not only to permafrost scientists but also to readers with a background in environmental science but without research expertise on permafrost. After an explanation of basic permafrost phenomena and processes, we describe the climate and climate change in the HKH. With this background, we go on to infer the distribution of permafrost in the HKH and the possible consequences of its thaw.

**Principles governing the occurrence and characteristics of permafrost**

Permafrost is relevant because it usually contains ice, has been frozen for long a time (e.g., thousands of years), and brings about strong geomorphic, hydrologic, and ecologic changes during thaw. While glaciers are not considered permafrost, landforms derived from glaciers and glacier beds can be subject to permafrost conditions. Seasonal freezing and thawing near the ground surface define the active layer with a typical thickness of about ½–8 m. Permafrost is an important element of landscape evolution owing to characteristic landforms such as rock glaciers, push-moraines, ice faces and hanging glaciers, and to its affects on long-term sediment transfer mechanisms. Unlike glaciers or snow, permafrost is usually concealed beneath the active layer and therefore precluded from effective observation with e.g., satellite-based remote sensing. Permafrost can be identified locally with direct measurements in boreholes, excavated pits, or with geophysical methods.

Permafrost and ground ice result from the long-term energy and mass exchange between the atmosphere and the subsurface. The fundamental processes are the same for permafrost in lowlands and in mountains (Haeberli et al., 2010; Harris et al.,

2009), whereas the configuration and relative importance of processes vary in differing environments. Regional climate, determined by global atmospheric and oceanic circulation as well as continentality and latitude, controls major patterns of air temperature, solar radiation, as well as the amount and seasonality of precipitation. Topography modifies these patterns through the barrier effect on precipitation, local thermal effects such as diurnal wind patterns and valley inversions, and the differentiation of solar radiation (Riseborough et al., 2008). Elevation strongly affects air temperature as well as short- and long wave radiation incident on the ground surface. In combination, these effects can cause differences in mean annual ground temperature of more than 10 ºC within a distance of less than one kilometer (Gruber et al., 2004b; Gubler et al., 2011; Hasler et al., 2011b).

The effects of climate and topography are further overprinted by ground conditions such as snow cover, vegetation, water availability, and subsurface characteristics, locally. Snow affects ground temperatures by providing insulation from the cold atmosphere when sufficiently thick, by its latent heat that needs to be overcome before ground beneath can warm above 0ºC, and by increasing albedo (Zhang, 2005). The net effect of these mechanisms, however, is highly variable and dependent on site conditions. Vegetation can lower ground temperatures by shading and evapotranspiration, and in other places exert a warming influence by retaining snow cover in wind exposed areas (Hinkel and Hurd, 2006; Kokelj et al., 2010). Ground material further differentiates subsurface temperature and some substrates such as an organic layer (Burn and Smith, 1988; Goodrich, 1982) or coarse blocks (Delaloye and Lambiel, 2005; Gruber and Hoelzle, 2008) result in a relative lowering of subsurface temperatures. Local drainage characteristics affect ground temperatures through latent heat and the effect of water saturation on thermal conductivity contrasts between frozen and thawed soil (Endrizzi et al., 2014) and through differing albedo of wet and dry soil (Ma et al., 1999). Subsurface ice such as snow or glacier ice buried in debris can act as a latent heat sink and lower ground temperatures. In total, local surface characteristics can cause a differentiation of several ºC in subsurface temperatures given the same conditions in climate and topography.

Subsurface ice largely determines the impacts of permafrost thaw and, like permafrost, escapes effective spatial observation (Mackay, 1972). The rapid freezing of moist sediment results in ice-cemented material with ice contents similar to the pore space in dry material. Material with higher ice content is referred to as 'ice rich' and originates from three main mechanisms: (A) Burial of snow or glacier ice below debris or moraine, resulting in diverse phenomena including buried snow (Gruber and Haeberli, 2009), rock glaciers (Haeberli et al., 2006), ice-cored moraines (Østrem and Ostrem, 1964), and debris-covered glaciers (Kirkbride, 2011). A wide variety of interactions and transitional forms exist between glaciers and permafrost (Etzelmüller and Hagen, 2005; Haeberli, 2005; Harris and Murton, 2005; Waller et al., 2012). (B) Ice segregation, resulting in progressive ice enrichment within permafrost (Guodong, 1983) that has even be observed in very arid mountain environments (Li and He, 1989). (C) Ice wedges, which on slopes are likely underestimated in their frequency of occurrence as their micro-topographic signature is suppressed by soil movement (Mackay, 1990). In bedrock permafrost, fractures often contain large amounts of ice, affecting slope stability and flow of water (Gruber and Haeberli, 2007; Ravanel et al., 2010), and the slow formation of ice-rich layers by segregation appears plausible (Girard et al., 2013). The freezing and melting of

ice in soil or rock occurs progressively over a range of temperatures below 0ºC and the high latent heat of fusion often subdues temperature change in frozen soil (Romanovsky and Osterkamp, 2000).

Because of this multitude of interacting factors, ground thermal regimes and ice contents resemble a highly variable mosaic, especially in mountains. This makes it difficult to predict permafrost conditions at individual locations, to evaluate computer models with a limited number of observations, to conclude from one feature (e.g., a rock glacier) the presence of permafrost in an adjacent area (e.g., a meadow), or to argue in terms of precise limits of permafrost occurrence. Our insight is, however, sufficient to describe important patterns of where permafrost is likely or unlikely to occur. On a coarser scale, the multi-decadal mean annual air temperature (MAAT) provides first estimates of the abundance of permafrost in an area (Brown and Pewe, 1973) and relationships long established for lowland permafrost appear to be adequate approximations in mountains (Boeckli et al., 2012). Often a MAAT of -1ºC is used to delineate areas with a considerable proportion of permafrost where local conditions are appropriate.

Besides air temperature, total precipitation is useful for conceptualizing expected spatial differences of permafrost occurrence (Haeberli, 2005). In locations with high precipitation, the equilibrium line on glaciers is situated at higher MAAT (lower elevation) than in more arid areas (Benn and Lehmkuhl, 2000; Yao et al., 2012). As a consequence, the proportion of glaciers covering slopes that could otherwise be underlain by permafrost is much larger in areas with high precipitation. Correspondingly, the areal proportion of permafrost in cold mountains with similar MAAT increases with decreasing precipitation. Furthermore, increasingly thick and insulating winter snow cover exerts a warming effect on the ground beneath it. Steep bedrock sheds its snow and can have permafrost in shaded slopes even at an MAAT near 0ºC (Boeckli et al., 2012; Hasler et al., 2011b), whereas sun-exposed slopes are without permafrost to much lower values of MAAT. Similarly, strongly wind-blown areas (Ishikawa et al., 2003) experience little warming by winter snow. Intact rock glaciers contain ice in the subsurface and are visible indicators of permafrost sometimes occurring at a MAAT as high as +2ºC. In continental and arid areas, permafrost can exist below a dense vegetation cover. This is possible because for plant growth summer temperatures are a limiting factor (cf. Jobbágy and Jackson, 2000; Körner and Paulsen, 2004) and in continental areas the high seasonal temperature amplitudes allow for warmer summers at the same MAAT than in less continental regions.

**Climate and climate change in the HKH**

Climate in the HKH is diverse due to the interplay of latitude, continentality, topography and atmospheric circulation. Most notably, the relative influence of winter Westerlies, the Indian and the East Asian monsoon in summer, as well as continental anticyclones over the QTP (Benn and Lehmkuhl, 2000; Böhner, 2006; Kapnick et al., 2014) differentiate precipitation amounts and patterns (Figure 1 CD). Precipitation in the central and eastern part of the Himalayas as well as parts of the QTP tends to be unimodal with a distinct Monsoon-related maximum in summer. In the western parts, the combined influence of Monsoon and Westerlies causes a bimodal precipitation pattern with precipitation both in winter and in summer

(Bookhagen and Burbank, 2010; Palazzi et al., 2013). In the far west of the HKH, a unimodal pattern of cyclonic winter precipitation can be found on the northern side of the Hindu Kush (Schiemann et al., 2008). The advection of differing air masses results in variable temperature lapse rates and, in some areas, seasonal precipitation regimes differing between arid valley bottoms, where most meteorological stations are situated, and adjacent high elevation areas with ample precipitation

(Hewitt, 2014; Immerzeel et al., 2015).

There is a paucity of climate stations in the HKH (Fowler and Archer, 2006; Salerno et al., 2015; Shea, 2015; Winiger et al., 2005), especially at high elevation. Furthermore, snowfall measurement is inherently difficult with errors often in the range of 20–50% undercatch (Rasmussen et al., 2012). Satellite-derived products such as the Tropical Rainfall Measuring Mission (TRMM) provide gridded precipitation in data-sparse regions but likely with reduced quality for snowfall and in high-

elevation terrain (Derin et al., 2016; Immerzeel et al., 2015; Yin et al., 2008). As a consequence, gridded climate data are poorly characterized in the HKH (Bao and Zhang, 2013; Ménégoz et al., 2013; Xie et al., 2007), especially at high elevation, and corresponding maps (including Figure 1 CD of this publication) need to be interpreted with care. This lack of reliable data, together with poorly known local phenomena and extreme elevation gradients, makes local prediction based on either the extrapolation of weather station data or on gridded atmospheric data challenging. These uncertainties invariably

propagate to model predictions of permafrost in the HKH.

Long-term observations in the HKH show atmospheric warming (Shrestha et al., 1999; Yang et al., 2011) exceeding global averages (Hartmann et al., 2013) and with pronounced regional differences in magnitude and timing. On the QTP (Liu et al., 2009) and in the central Himalaya at 2660–5600 m a.s.l. (Salerno et al., 2015) diurnal minima were observed to increase faster than maximum temperatures. While this is similar to observations in other mountain ranges such as the European Alps

or the Rocky Mountains, a dominant increase of diurnal maxima was observed in the western Himalaya (Bhutiyani et al., 2007; Shekhar et al., 2010) and on the southern slope of the central Himalayas between 1300–2500 m a.s.l. (Kattel and Yao, 2013). Seasonally, the strongest warming has been observed during winter months in the northwest (Bhutiyani et al., 2007) and central (Salerno et al., 2015) Himalaya and on the eastern QTP (Liu et al., 2009). Often, stronger warming is found at high elevation (Liu et al., 2009) and this trend has been attributed to feedbacks related to snow and cloud cover.

Precipitation regimes in the HKH (Figure 1 CD) as well as their changes are diverse. Most notably, several studies detect a recent weakening of the Monsoon (Palazzi et al., 2013) with a significant reduction in liquid precipitation at high elevation in the central Himalaya (Salerno et al., 2015). Monsoon precipitation is strongest in areas of high relief and mid elevation (Bookhagen and Burbank, 2006, 2010), above this a sharp decrease of precipitation with elevation has been shown in the central Himalaya since the mid 1990s (Salerno et al., 2015, 2016). Climate model scenarios include both increasing and

decreasing monsoon intensity (Moors et al., 2011; Palazzi et al., 2013) and thus an element of uncertainty even with respect to the direction of future changes. For the QTP, an increase in precipitation has been inferred between about 1960 and 1990 (Liu et al., 2011) and between 1951 and 2000 an increase in winter precipitation together with a partial decrease in summer precipitation were found (Zhai et al., 2005).

These observations underscore the notion that future climate change will be more pronounced at high elevations (Pepin et al., 2015) in the HKH than on a global or hemispheric average. The spatial and seasonal differences in the trends of air temperature and precipitation will compound difficulties in predicting permafrost based on gridded atmospheric data.

Snow cover in the HKH, driven by precipitation and temperature, exhibits a similarly strong variability with elevation, season, and region as evidenced by a number of studies. Stable or slightly increasing snow cover, contrary to intuitive notions of global warming, is reported in the Karakoram during 2000–2009 (Tahir et al., 2011). Similarly, an increasing trend in snow cover was found for the QTP for the period 1951–1997 (Qin et al., 2006) but a slight decrease was detected for 2000–2006 (Pu and Xu, 2009). Snow cover in the interior of the QTP tends to be thin, patchy and of short duration with large inter-annual variability (Pu and Xu, 2009; Qin et al., 2006). On the QTP, snow cover is more persistent along the southern and western margins in high mountains and the seasonal cycle is differentiated with elevation (Pu and Xu, 2009): Below about 5000 m a.s.l., snow cover fraction is high in winter and low in summer, whereas at higher elevation, two peaks in snow cover fraction are detected in late October and early May with a second relative minimum December and January that is likely related to sublimation, snow redistribution, and reduced precipitation.

Regional glacier mean elevations range from 5150 m asl in the west to 5600 m asl in the central HKH (Bolch et al., 2012) and equilibrium line altitudes (ELA) from 4400 to 5900 m asl (Yao et al., 2012). The derivation and interpretation of these metrics, however, is subject to some uncertainty (Benn and Lehmkuhl, 2000). Observation of glacier changes, similar to atmospheric measurements, points to an overall warming trend superimposed with large regional differences (Kääb et al., 2012). Most glaciers have retreated over much of the HKH in recent decades (Kääb et al., 2012; Yao et al., 2012), with increasing loss since about 1995 (Bolch et al., 2012). The local observation of climate and glaciers thus agree with the global trend towards atmospheric warming, which is likely to accelerate in the coming decades.

**Occurrence and characteristics of permafrost in mountains of the HKH**

Early reviews (Brown et al., 1998; Gorbunov, 1978; Matsuoka, 2003) already predicted large areas of permafrost in the HKH based on insight from neighboring mountain ranges and limited local field evidence. The areas of likely permafrost occurrence in mountains can now be more consistently constrained and visualized (Figure 2) based on a simple global model (Gruber, 2012) with about 1km spatial resolution. In a first-order evaluation, this approximate simulation has been corroborated in the HKH with remote rock-glacier mapping (Schmid et al., 2015). Based on this model, which contains an estimate of uncertainty, the overall permafrost area is about one million $km^2$ or 14 times the area of glaciers in the HKH at present (Table 1). With two thirds of the QTP underlain by permafrost, China has by far the largest share of this. With the exception of Bhutan, the expected permafrost area is significantly larger than the glacier area in all countries (Table 1, Figure 3). The estimated elevation range of permafrost occurrence is large due to the regional variation in air temperature and topography. In some areas in the West of the HKH, permafrost can be expected to occur as low as 3500 m a.s.l. At the same

time, bedrock without permafrost is likely to exist in south-exposed faces to well over 6000 m a.s.l. in the central and eastern part of the HKH.

Rock glaciers are indicators of permafrost in mountains, their identification and the visual assessment of their activity (ice content) however, are often ambiguous (Hewitt, 2014; Schmid et al., 2015). Rock glaciers have been described in several areas of the HKH (Barsch and Jakob, 1998; Jakob, 1992; Regmi, 2008) at 3500–5500 m a.s.l. (Schmid et al., 2015). Their origin was interpreted as related to either talus slopes or glaciers and their moraines (Hewitt, 2014; Ishikawa et al., 2001; Owen and England, 1998; Shroder et al., 2000). The importance of rock glaciers in terms of abundance, sediment transport, and in advancing into modern floodplains has been described for the Lahul and Garhwal Himalayas and the Karakoram (Hewitt, 2014; Owen and England, 1998). While low precipitation has been recognized as favoring the development of rock glaciers, many examples have also been documented in humid parts of the HKH (Hewitt, 2014) and an absence of rock glaciers in extremely arid regions has been noted (Fort and van Vliet-Lanoe, 2007). Rock glacier complexes (Hewitt, 2014) consisting of tributaries coalescing into lobes 2–3 km wide and terminating in multiple lobes of sometimes differing activity and velocity have been described in the Karakoram. There, rock glaciers affect hydrology, mountain pastures, and tree cover in headwater basins. They impound lakes and frequently have small streams of clear water and reliable discharge associated with them (Hewitt, 2014). One author stated that rock glaciers seem to be more directly and extensively part of human settlement and land use in the Karakoram than glaciers are (Hewitt, 2014). The origin of ice-debris landforms in the HKH is often not clearly attributable to either glacial or periglacial processes.

Cold ice can sometimes be visually identified and provides indications of permafrost in contact with it. Other than temperate ice, which is at the melting point, cold ice usually does not contain much liquid water. Hanging glaciers and ice faces, visible in many iconic peaks of the HKH, are only stable in steep slopes because they are frozen to the underlying rock (Gruber and Haeberli, 2009). In arid regions, many small glaciers have cold tongues, which often are identifiable from their supraglacial drainage channels (Ryser et al., 2013). Indicators such as rock glaciers and hanging glaciers are reminders of the existence of permafrost in a landscape but have limited value for inferring the existence of permafrost beneath other surface features nearby.

Direct observations of permafrost in mountains of the HKH are sparse. The majority of Chinese permafrost research is focused on the QTP engineering corridor (Yang et al., 2010a) and observations in mountains aimed at understanding the effect of topography are rare by comparison. Nevertheless, unique insight into arid lands permafrost is provided by Chinese research and is invaluable in estimating permafrost conditions in the HKH. Outside of China, only a handful of ground temperature measurements (Fujii and Higuchi, 1976; Klimeš and Doležal, 2010; Shiraiwa, 1992) and few published observations of ground ice exist (Rastogi and Narayan, 1999). Nevertheless, knowledge on permafrost processes and on climate conditions in the HKH support a number of inferences about the characteristics and spatial distribution of permafrost in mountains of the HKH. The diverse and extreme character of the region implies a wide range of permafrost characteristics that can be expected to exceed what is known from other environments. That is why this large-area review can only infer

possible processes and phenomena for broad ranges of environments based on existing research, and concrete studies or reviews will require a more narrowly defined and quantified set of environmental conditions.

Permafrost likely follows known patterns in many ways: given the same MAAT, a larger abundance of permafrost is expected in arid areas than in areas with more precipitation. In humid areas, few debris slopes are expected to have permafrost with the exception of windblown areas devoid of snow, coarse block covers, peat lands, or debris with incorporated meteoric ice. In more arid areas, extensive permafrost occurrence is to be expected, even below vegetated surfaces. The distribution of permafrost in steep bedrock is much less affected by precipitation as sliding controls snow amounts. The effect of monsoonal precipitation regimes with wet summers and dry winters (Figure 1D) on the differentiation of ground temperature with respect to MAAT is unknown. Possible effects can include reversed thermal offset (Lin et al., 2015) as well as a changed net effect of snow cover, providing little insulation in winter and frequent increases in albedo and latent heat in summer.

Direct solar irradiance, which strongly controls the spatial differentiation of ground temperature in mountains, increases with elevation. By contrast, spatially more uniform fluxes (diffuse solar radiation, thermal irradiance, turbulent heat transport) tend to decrease (Gruber et al., 2004b). As a consequence, the temperature difference between shaded and sun-exposed slopes at extreme elevations likely exceeds known values.

Extreme aridity subdues typical periglacial forms (Fort and van Vliet-Lanoe, 2007; Wang and French, 1995). As a consequence, the geomorphic surface forms typically associated with permafrost may be absent in parts of the HKH. Aridity, and in more humid conditions topography, give rise to unsaturated conditions in the subsurface and alter the heat and moisture transport in the soil. In arid areas, ice-rich permafrost is known to exist (Zhang et al., 2008) and its amount may be governed by topography (Wang and French, 1995). It is conceivable that the incorporation of moisture into the top of permafrost (Figure 4) provides a form of water storage capable of accumulating in more cold and humid periods and delayed release during unusually warm conditions. Many small lakes, often with signs of thermokarst, have been documented on the QTP and occur predominantly in ice-rich ground (Lin et al., 2011; Niu et al., 2011).

The burial of snow or avalanche deposits (Gruber and Haeberli, 2009) in talus slopes and cones is expected to cause large aggrading bodies of permafrost. Catchment size, topography, temperature, and relative ice content govern whether these mixtures develop into landforms with little movement, into rock glaciers, or into debris-covered glaciers. While moving landforms can be discerned visually, rather stationary ones can contain ground ice without easily visible clues. Buried ice can occur near the margins of present-day glaciers and in formerly glacierized areas. When this occurs in cold permafrost, buried ice can persist for long times and thus allow surface processes to gradually obliterate surface expressions of buried ice. Strong winter cooling can furthermore support local aggradation by surface melt water refreezing in soil macro-pores or in snow accumulations generated by wind drifting or avalanches (superimposed ice). Corresponding bodies of ice are visible on the surface and in the shallow subsurface (Figure 4A–F) and can be expected to exist more frequently at depth. Large areas with ground ice are to be expected in the HKH and depending on ground temperature, which is indicative of the potential to endure warm periods, this ice can be very old.

The QTP and greater Himalayas had similar elevation as today and were subject to varying patterns of monsoonal circulation throughout the last 2–4 million years (Clift et al., 2008; Zhisheng et al., 2001). Pleistocene maximum glacier extent occurred at times different from global maxima and, in response to changing circulation patterns, exhibited large regional differences (Benn and Owen, 1998). During the Holocene, a climate optimum associated with warmer and/or wetter conditions has been
inferred (Benn and Owen, 1998). Such patterns of changing precipitation and temperature point to spatially variable ages, depths, and ice contents of permafrost.

In steep terrain, three-dimensional effects (Noetzli et al., 2007; Noetzli and Gruber, 2009) can induce permafrost beneath slopes with positive mean annual ground temperatures due to heat conduction from nearby colder slopes. Similarly, temperate firn areas within otherwise cold-based glaciers may cause localized taliks in permafrost. Many of the high HKH
peaks have complicated three-dimensional permafrost bodies inside.

In summary, some of the distribution and characteristics of permafrost in the mountainous parts of the HKH can be anticipated. At the same time, differing combinations of high elevation, monsoonal precipitation regimes, and aridity are likely to bring about a wealth of new and unexpected phenomena.

**Persistence and impacts of permafrost thaw in the HKH**

In the past two decades, the atmosphere has warmed throughout most of the HKH as observable in the accelerated loss of surface ice. Subsurface ice contained in permafrost, which is not readily observable, is expected to undergo changes of similar extent as the energy balance at the ground surface similarly drives it. Climate change (air temperature, precipitation, cloudiness) in the HKH exhibits diverse regional patterns as well as differing trends at high/low elevation or during differing seasons. Some speculation on the effect of the diverse regional changes can inform future research: Winter warming, for
instance, may have a weaker effect on ground temperatures than warming during the summer months due to the insulation by snow. Short-lived climate pollutants such as black carbon, dust, and aerosols can shorten the duration of the snow cover (Ménégoz et al., 2014) and thus exert a warming effect on the ground by exposing it earlier to warming surface fluxes. A delayed onset of the snow cover in areas subject to winter westerlies may promote increased ground cooling. As these changes are further overprinted by topography, the resulting effect on permafrost is likely to be spatially highly diverse as
well. With intensifying climate change, widespread and persistent thaw of permafrost in the HKH is therefore likely in the decades to come. This is corroborated by observations of increasing permafrost thaw in diverse permafrost areas globally (Romanovsky et al., 2010) and on the QTP (Shaoling et al., 2000).

The energy exchange between the atmosphere and the subsurface is moderated by interacting factors such as air temperature, precipitation amounts and seasonality (monsoon), insolation, snow cover, surface material, vegetation cover, and subsurface
properties. For this reason, differing facets of a landscape often differ drastically in their response to the same change in climate (Ma et al., 2015; Riseborough et al., 2008). Surface effects such as changes in vegetation or snow cover can thereby amplify or counteract atmospheric forcing. To thaw permafrost, heat needs to be transferred between the ground surface and

the permafrost at depth. As this is dominated by heat conduction, thaw is increasingly delayed with depth. Ice contained in the subsurface further slows the thaw of permafrost by absorbing heat during melt – the energy input required for changing ice to water is equivalent to raising the temperature of an equal volume of rock by about 150 ºC (Gold and Lachenbruch, 1973). As a consequence, subsurface ice content affects both the impacts and the persistence of permafrost, which can easily

span decades to centuries for thicknesses of tens to hundreds of meters. In summary, the highly variable mosaic of permafrost properties in mountains is further complicated by diverse trajectories of change imposed by variable surface and subsurface conditions. Therefore, mapping and forecasting impacts of permafrost thaw is difficult. Nevertheless, sustained atmospheric warming is expected to increase subsurface temperatures globally and this will be accompanied by at least partial thaw into the often ice-rich upper layers of nearly all permafrost.

The simplest impact of thawing ice-rich permafrost is ground subsidence due to ice loss. Additionally, changing mechanical properties (Arenson et al., 2007) can affect the rates, magnitudes, and locations of sediment removal by creep, sliding, or by flowing water. Ice increases the mechanical strength of sediments by providing cohesion. When frozen sediments thaw, their mechanical properties change drastically. Less obviously, even a warming of frozen ground at temperatures below zero can influence its mechanical stability via temperature-dependent properties of ice and changing liquid water contents (Arenson et

al., 2007; Kurylyk and Watanabe, 2013). Similarly, warming is hypothesized to cause acceleration or destabilization of rock glaciers (Buchli et al., 2013; Kääb et al., 2007). The fast thaw into ice-rich fine material, which often originates from ice enrichment in the top of permafrost, can cause slope failures at surprising low angles (McRoberts and Morgenstern, 1974). Where massive ground ice is exposed in a cut slope, retrogressive thaw slumping can develop (Niu et al., 2012; Wang and French, 1995). The thaw of ice in aggrading slopes (Gruber and Haeberli, 2009) can expose previously accumulated stores of

sediment to erosion by shallow landsliding and debris flows.

Warming and thaw of ice-filled rock clefts result in a transient reduction of stability that can trigger rock slope failure, increased triggering during earthquakes (Kargel et al., 2015), and lead to an acceleration of large-scale deformation. This is further enhanced by the potential of thawing clefts to build up hydrostatic pressure (Gruber and Haeberli, 2007). The role of permafrost in translating heat waves into rock fall is supported by field observations (Gruber et al., 2004a; Ravanel et al.,

2010). The fast reaction of rock fall to summer heat waves (Gruber et al., 2004a) bears witness to a fast connection between surface conditions and stability conditions at depth. While the conductive warming of convex topography is faster than in flat terrain (Noetzli and Gruber, 2009), this is likely caused by either heat advection by water percolation (Hasler et al., 2011a) or by thermo-mechanical forcing (Hasler et al., 2012). Changing surface ice cover can affect the permafrost below it; instability in steep glaciers and in permafrost bedrock has been observed to occur together and to produce large ice-rock

avalanches (Haeberli et al., 2004; Huggel et al., 2008; Lipovsky et al., 2008). In summary, permafrost thaw results in an increased frequency and possibly in unexpected locations of mass movements, as well as in increased sediment loads available for further downstream transport.

Mass movements in mountain terrain often have a far reach due to their high potential energy. This can be further extended due to the influence of ice content in the moving mass or the trajectory running over glacier ice (Schneider et al., 2011).

Furthermore, landslides in mountain environments typically are complex, i.e. one mode of movement transforms into another. Possible transformations include rock slides turning into debris avalanches (Geertsema et al., 2006a) and debris flows (Geertsema et al., 2006b), and deposits of mass movements acting as dams creating temporary lakes. The sudden drainage of lakes can cause flooding and large debris flows (Osti and Egashira, 2009; Vuichard and Zimmermann, 1987). In

this context, permafrost is relevant in terms of melting ice cores or ice-cement (Fujita et al., 2013; Watanabe et al., 1995) within the dam and in terms of triggering a breach when fast mass movements enter lakes and their displacement waves contribute to overtopping of dams. Large glacial lakes (cf. Figure 2C) are found within and outside the zone where recent permafrost formation is expected (Iwata et al., 2004) and often dammed by debris-covered stagnant ice bodies (Richardson and Reynolds, 2000). In a first glacial lake inventory for Bhutan, Nepal and selected areas in China, India and Pakistan, 8790

glacial lakes have been identified and 203 classified as potentially dangerous (Ives et al., 2010). Glacier recession is expected to result in a further increase of their number and often also their size (Gardelle et al., 2011). The presence of steep bedrock permafrost above natural lakes, and possibly man-made reservoirs, gives the potential for increased frequency of large rock fall a much wider practical relevance. The extreme size of some glacial lakes in the HKH and correspondingly large outburst volumes give these events a potential for having extreme run-out distances (Richardson and Reynolds, 2000)

far beyond the area of permafrost occurrence. Often, processes related to glaciers and to permafrost conspire in producing high-mountain hazards. These represent a continuous threat to human lives and infrastructure and related disasters can kill thousands of people at once and cause damage on the order of 100 million dollars per year, globally (Kääb et al., 2005).

Rock glaciers are creeping landforms consisting of frozen debris. A broad range of origins exists including debris-covered cold or polythermal glaciers (Bolch and Gorbunov, 2014), ice-cored moraines, and forms occurring independent of glaciers

in their direct vicinity (Haeberli et al., 2006). Their typical velocities are in the range of a few cm to a few meters per year. A number of rock glaciers with velocities of several ten meters per year, with signs of disintegration (Roer et al., 2008), and one complete collapse (Krysiecki et al., 2008) of a rock glacier have been documented outside the HKH. These phenomena are hypothesized to be driven by a decadal-scale warming trend (Delaloye et al., 2008; Kääb et al., 2007; Sorg et al., 2015) and to react to seasonal forcing (Wirz et al., 2016). A widespread acceleration of rock glaciers can affect geohazards by

supplying material to debris-flow starting zones and as a mobilization of large sediment stores (Otto et al., 2009; Owen and England, 1998), resulting in increased material delivery to streams (Gärtner-Roer, 2012). Rock glaciers, within which ice melt is retarded by thick debris, can thus advance in times when glaciers retreat. Meltwater release from rock glaciers and their role as aquifers affect local hydrology (Burger et al., 1999; Hewitt, 2014). Especially in arid mountain environments, they are considered an important element of hydrology (Azócar and Brenning, 2010; Rangecroft et al., 2015).

Permafrost thaw affects hydrology through increasing hydrologic permeability with ice loss in pores and through the release of water stored in frozen material. As an aquitard, permafrost can support perched water tables and increase near-surface soil moisture (Gorbunov, 1978; Ishikawa et al., 2005). In arid areas, wetlands often occur where this shallow groundwater seepage is focused by topographic depressions (Woo et al., 2008). Permafrost also enables artesian groundwater, where groundwater recharge or discharge (springs) can only occur in localized taliks. Permafrost thaw, through talik growth and

higher permeability of partially frozen material, changes the hydrologic connectivity (Hinzman et al., 2005) in a catchment and affects the rates and amounts of flow along differing paths. An increasing proportion of deeper ground water flow has been postulated based on observations and computer simulation (Bense et al., 2012; Frey and McClelland, 2009) and linked with altered temporal runoff characteristics (Woo et al., 2008) such as higher base flow. Similarly, an observed strong

lowering of groundwater and lake levels has been linked with permafrost thaw (Cheng and Jin, 2012; Jin et al., 2009). The thaw of ground ice releases water from long-term storage into the annual hydrologic cycle. As ground ice content, on average, is highest near the top of permafrost this can result in a transient pulse of water release under sustained atmospheric warming. This effect, however, will eventually diminish as thaw progresses into deeper layers with less excess ice and stronger retardation by heat diffusion, and into progressively smaller areas of the regional hypsometry. The magnitude and

timing of this effect will be difficult to predict (Cheng and Jin, 2012) and vary regionally and with the type of ground ice.

Water chemistry is affected by permafrost thaw via changed flow paths and the release of solutes from newly thawed material. Differing flow paths result in distinct geochemical signatures (Clark et al., 2001). As the hydrologic regime changes from near-surface drainage to deeper flow paths following active layer deepening and talik formation, changed cycle times and their proportional contribution to stream flow will affect solute contents in surface water (Frey and

McClelland, 2009; Hinzman et al., 2005) and groundwater (Cheng and Jin, 2012). Additionally, the thaw of previously frozen material releases solutes (Hinzman et al., 2005; Kokelj and Burn, 2003; Williams et al., 2006). This can be due to the release of ions stored in ground ice (Kokelj et al., 2002; Kokelj and Burn, 2003) or, on longer time scales, caused by exposure of new material to leaching, as previously frozen deposits (Gruber and Haeberli, 2009) become hydraulically more permeable. In some cases, water originating from areas of inferred permafrost degradation has been shown to contribute to

exceeding guideline values for drinking water quality with respect to selected heavy metals (Thies et al., 2007, 2013). Based on the dilution with direct runoff and water from non-permafrost areas, these effects on water quality are likely most pronounced in small and arid catchments.

Moisture conditions and the release of solutes and organic matter previously immobilized in frozen material affect vegetation and biochemical cycling. In arid areas, increased soil moisture due to perched groundwater promotes plant growth

(Gorbunov, 1978) and in some areas, a clear relation between wetlands and permafrost is established (Jin et al., 2008b, 2009; Yang et al., 2010b). Such wetlands have been shown to dry up and evolve into semi-deserts and steppes following permafrost degradation (Cheng and Jin, 2012). Other desertification processes and loss of vegetation often result in soil warming and hence impacts on permafrost (Yang et al., 2010a). Similarly, a changed location of creeks seeping from water flowing over a permafrost aquitard was reported to cause the degradation and desertification of wetlands and pastoral land

(Jin et al., 2009). Thaw-related mobilization of carbon stored in permafrost (Schuur et al., 2015) releases greenhouse gasses. In corresponding global estimates of carbon pools, little information exists on mid or low latitude mountain regions, which are instead considered with simple factors only (Zimov et al., 2006). Valley fills (Blöthe and Korup, 2013), deposits of solifluction, and aeolian material are environments in which carbon may have been accumulated and preserved in permafrost

over long time scales, even in arid environments, although likely of smaller magnitude than high-latitude carbon storage (Fuchs et al., 2015).

With respect to water and ecosystems, the response of permafrost to climate change will have some similarity to that of glaciers but will be more delayed and persistent: an transient increase in runoff during warm periods followed, after years or decades, by a reduced capacity to buffer stream flow by water release in warm periods. The relative magnitude of impacts related to runoff, water quality, and vegetation will be greater in arid areas where the areal abundance and water content of permafrost is significant compared to that of glaciers. As the reaction of snow cover duration and glacier recession will be faster than permafrost thaw at depth, the relative proportion of runoff derived from thawing subsurface ice will increase further. A transient increase in vegetation abundance can be imagined to result from the release of water and nutrients during sustained permafrost thaw, but over time will likely diminish into a permanently dryer and less vegetated state in a landscape transitioning to a smaller proportion of near-surface runoff.

The engineering properties of frozen ground are strongly dependent on temperature. While permafrost a few degrees below 0ºC mostly has high strength in compression, excellent bearing capacity, and low hydraulic permeability, these properties change drastically when thaw occurs and ground ice gradually turns to water. Permafrost beneath engineered structures will be influenced by not only the thermal disturbance of the structure, but also the thermal impact of long-term climate change. As a consequence, engineered structures such as dams, roads, railways, or buildings will require increasing maintenance and repair as permafrost thaws. Similarly, geohazards affecting such structures can change their frequency of occurrence or occur from location previously considered safe. Frost mounds (Jin et al., 2008a), migrating pingos (Wu et al., 2005), or landslides (Huggel et al., 2013; Kargel et al., 2015) are examples of geohazards conditioned by permafrost. The consequences of permafrost thaw on structures can be mitigated by careful site selection and specialized building practices (Bommer et al., 2010; Zhang et al., 2008). Permafrost engineering has a long tradition based on development and infrastructure operation in Russia, North America, China, and Northern Europe. In HKH countries other than China, however, little awareness and expertise exists with respect to the design of structures for permafrost conditions or the assessment of possible geohazards connected with permafrost thaw. At the same time, infrastructure on or near permafrost exists in those countries (Figure 2).

**Perspectives**

Large areas of permafrost exist in the mountainous parts of the HKH. Despite its extent and likely scientific and societal relevance this permafrost has hardly been investigated. Based mostly on research from other regions or continents, we conclude that widespread permafrost thaw is virtually certain to occur in the future and that this persistent change in subsurface conditions will affect sediment budgets and geohazards, hydrology, water quality, and vegetation. The importance of permafrost phenomena in a region or catchment will be locally variable much like the impacts of glacier melt have differing magnitudes in headwater basins and in major rivers hundreds of kilometers downstream. The full diversity of impacts as well as pathways connecting them with human activity, however, are largely unknown because environmental

and societal conditions in the HKH are markedly different from other permafrost areas and because relevant observations, data, and regional expertise are sparse. In this context, coordinated research and measurement programs are required, both to gain confidence in the application of knowledge gained in other regions, and to investigate phenomena unique to the HKH. Such research programs will strengthen local expertise and networks and provide important links to the international research community. Long-term monitoring of ground temperature, ice content, and other variables at selected sites (cf., Vonder Mühll et al., 2008) will contribute to national and international programs and provide a basis for the evaluation of simulation and remote sensing products.

For the basic understanding and assessment of permafrost environments a number of broad research questions emerge: What types of permafrost terrain are relevant and how can they be recognized? What amount and form of ground ice exists beneath differing terrain types? How fast is permafrost warming and losing ice in differing climates and terrain types? How can ice-debris landforms be better understood in combined permafrost and glacier studies? How do regional climates such as monsoon, high elevation, or aridity affect permafrost? What is the role of permafrost and seasonally frozen ground in controlling discharge and water quality in arid areas?

Similarly, understanding the physical impacts of permafrost thaw raises a number of questions: How much water is stored in frozen ground? How is melting ground ice affecting water resources in terms of quantity, timing, temperature, chemistry, sediment load, and freshwater ecology? What will be the long-term implications for local freshwater supply? What are the regional relationships between vegetation or wetlands and permafrost characteristics? What is the areal extent of pastures and rangelands in permafrost conditions? What are regionally important geohazards related to permafrost and its thaw? How do cascading processes connect permafrost and populated places? Is a changed geohazard regime detectable in permafrost areas? The answers to many of these questions may differ from established insight elsewhere, and likely locally, within the HKH.

The diversity of cultures and livelihoods present at high elevation in the HKH also motivates questions on the local interaction of humans and permafrost: How are permafrost phenomena and their change represented in local knowledge? Where in the HKH are significant amounts of people living on or close to permafrost? What are possible socio-economic and cultural consequences of permafrost thaw? To what extent do agro-pastoralists depend on pastures and rangelands in permafrost? Which (high value) species are linked with permafrost and may disappear with its thaw? What are suitable adaptation strategies to mitigate the possible impacts of permafrost thaw on local populations? How are research findings best communicated to the affected population?

Given the size and diversity of the HKH and the remoteness of many populated places, the ability to simulate permafrost conditions and to use remote sensing methods for the detection of relevant changes and impacts is important. This is because local measurements will not suffice to produce the first-order maps of permafrost characteristics or change needed. Computer simulations help in distinguishing plausible from less likely future developments. Promising methodologies for improved simulation in remote locations and mountain areas exist (Fiddes et al., 2015; Fiddes and Gruber, 2012, 2014; Westermann et al., 2015) and offer synergies with efforts in atmospheric sciences (Gutmann et al., 2016; Ménégoz et al.,

2013), glaciology, and hydrology. Furthermore, the trend to using more physics-based models, towards more rigorous model testing (Gubler et al., 2013), and towards greater attention to scaling issues (Muster et al., 2012; Zhang, 2013) will likely result in a more unified approach to simulating permafrost in both polar or sub-polar lowlands and lower-latitude mountain areas. Simulation and remote sensing will be key tools in the assessment of permafrost moderated climate impacts and related risk reduction.

Several recent initiatives aim at investigating permafrost in mountainous areas of the HKH. Most prominently, ICIMOD has lead and supported a number of activities since 2013, both in Nepal and in other HKH countries. ICIMOD researchers have installed more than 25 near-surface ground temperature loggers in Langtang Valley, Nepal between 4500 and 5500 m a.s.l. in 2013 and 2014 with data collection/service in 2015 and 2016. In 2016, 14 additional loggers were installed here by a project from the University of Zurich and the WSL Institute for Snow and Avalanche Research SLF, Switzerland, in collaboration with ICIMOD, Kathmandu and Tribhuvan University, Nepal. The Glacier Trust initiated the HiPer Summer School with fieldtrip in the Khumbu area in 2014. It is focused on periglacial research and capacity building, collaborating with Tribhuvan University and ICIMOD. The Indo-Swiss programme IHCAP (Indian Himalayas Climate Adaptation Programme) has resulted in maps of estimated permafrost distribution for the Kullu district, Himachal Pradesh, India (Allen et al., 2016). A research project aimed at the characterization of glacial lake deposits for geomorphic and climatic changes at BSIP, Lucknow, India has sampled permafrost (>80 cm deep) in the sediment of palaeolake deposits in Ladakh in June 2015 (Figure 4. G-I). In August 2016, an international workshop 'Himalayan Permafrost under the Changing Climate' was held in Delhi. Afterwards, scientists from the Indian National Institute for Hydrology, ICIMOD, and Carleton University, Canada described 13 plots in a high-elevation research catchment near Leh, India, and instrumented them with two near-surface ground temperature loggers, each. During this campaign, observations on near-surface ground-ice were made (Figure 4A–F). In October 2016, ICIMOD and Karakoram International University, Gilgit, Pakistan, in collaboration with Carleton University, conducted a Summer School on Permafrost in Islamabad, Pakistan. During this training, the installation of temperature data loggers as well as the gathering of other relevant observations in the Karakoram were prepared. This list, which is likely not complete, shows a clear trend in growing engagement and capacity for permafrost research in the HKH.

The importance of permafrost for many phenomena is as easy to overlook as it is to not see permafrost itself: Landslides exist without permafrost. Engineering challenges exist without permafrost. In regions without permafrost, climate change brings about hydrologic and ecologic change. The importance of permafrost lies in the profound transformation of physical characteristics of the subsurface in response to temperature change and the additional element of change and surprise this brings for the systems concerned. As many domains of environmental research (e.g., glaciology, hydrology, climatology) face similar challenges, there is great potential for synergy in joint research and discovery in the HKH.

**Author contributions**

S. Gruber wrote most of the manuscript and produced the figures and tables. Additional input on hydrologic impacts was provided by P. Panday, on permafrost and lakes by M.O. Schmid, on permafrost engineering and permafrost on the QTP by E. Guegan, and on wetlands by R. Fleiner. L. Zhao and Y. Zhang contributed expertise regarding permafrost on the QTP and in mountains. S. Gruber, D. Stumm, and P. Wester conceived and planned the manuscript and review activities, and D. Stumm, and P. Wester contributed to the writing, commenting and editing of the manuscript. The views and interpretations in this publication are those of the authors and they are not necessarily attributable to their organizations.

**Acknowledgements**

The International Centre for Integrated Mountain Development (ICIMOD), Nepal, provided funding and logistical support for this review. This study was partially supported by core funds of ICIMOD contributed by the governments of Afghanistan, Australia, Austria, Bangladesh, Bhutan, China, India, Myanmar, Nepal, Norway, Pakistan, Switzerland, and the United Kingdom. The future research directions outlined in this contribution are partially inspired by the outcomes of a group discussion during the International Symposium on Glaciology in High Mountain Asia, Kathmandu, Nepal, in March 2015. Dr. R. Thayyen provided invaluable guidance during the 2016 field excursion that resulted in the images shown in Figure 4 A–F. Dr. F. Salerno and two anonymous reviewers provided valuable feedback and input during the discussion stage of this manuscript.

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

**Tables**

**Table 1:** Comparison of permafrost and glacier areas. Permafrost area is given as a best guess with an estimated uncertainty range in brackets (Gruber, 2012). Glacier area is derived from differing inventories resulting in two numbers per country. The first glacier area given is based on an inventory for around the year 2005 (Bajracharya and Shrestha, 2011). For China, this inventory is lacking parts of the 5 Karakoram and Pamir. The second glacier area is based on the Randolph Glacier Inventory 5.0 (Pfeffer et al., 2014). For comparison, the permafrost area in the entire European Alps is about $6 \times 10^3$ km$^2$.

| Country | Permafrost area in HKH [$10^3$ km$^2$] | Glacier area in HKH [$10^3$ km$^2$] |
|---|---|---|
| **Afghanistan** | 17.5 (5.9–31.6) | 2.6 / 3.1 |
| **Bhutan** | 1.2 (0.3–3.5) | 1.0 / 1.5 |
| **China** | 906 (486–1356) | 29.2 / 33.6 |
| **India** | 40.1 (10.5–77.4) | 11.8 / 19.6 |
| **Myanmar** | 0.1 (0.03–0.5) | 0.04 / 0.06 |
| **Nepal** | 11.1 (6.4–16.1) | 4.1 / 4.8 |
| **Pakistan** | 26.6 (7.2–53.3) | 11.3 / 17.0 |
| **Total** | 1003 (516–1538) | 60.1 / 79.8 |

**Figure 1**

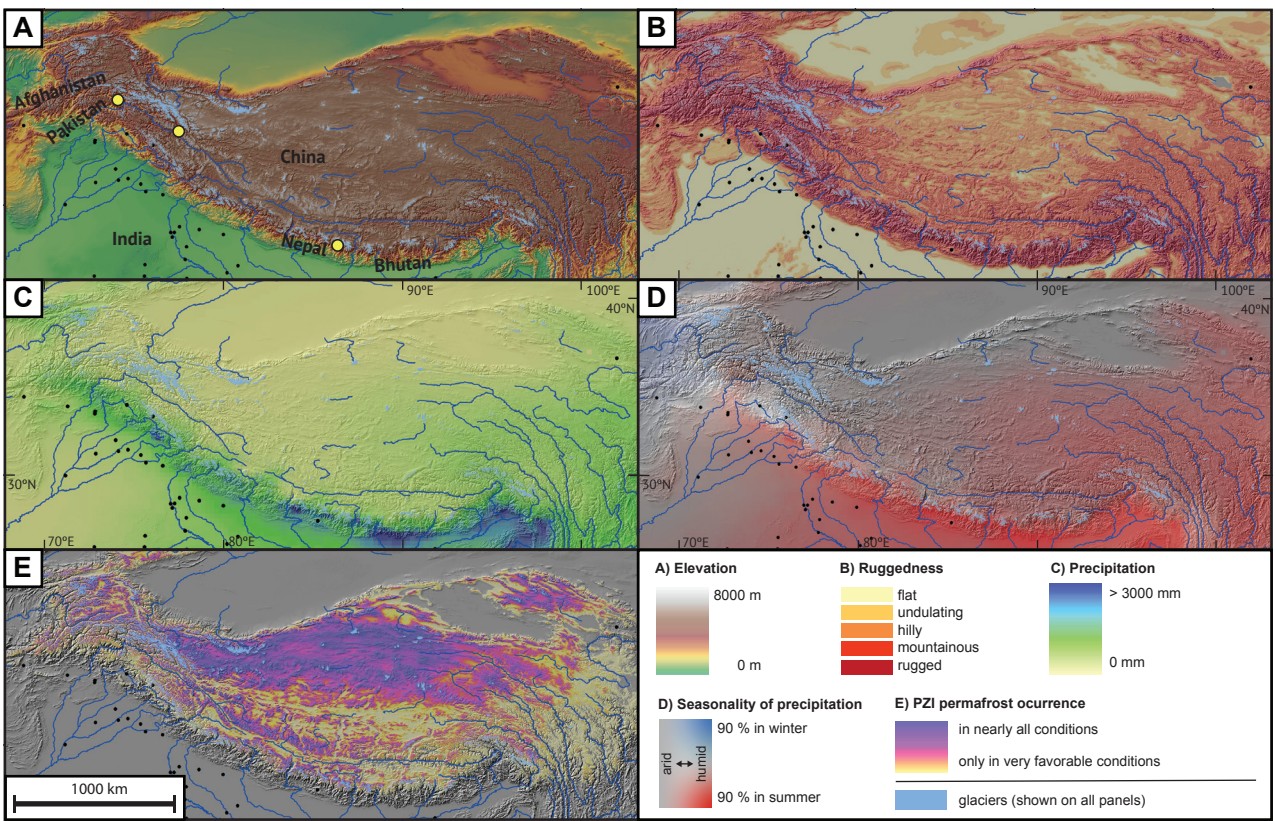

**Figure 1:** HKH region: (A) Physiography, yellow dots indicate locations shown in Figure 2. (B) Terrain ruggedness. (C) Mean annual precipitation sum 1988–2007. (D) Seasonality of precipitation expressed as the percentage received during winter (October to March) or summer (April to September) for 1988–2007. Where seasonality is unclear due to equal amounts or low total precipitation no color signature is shown. (E) Permafrost zonation. Urban areas and major rivers are shown for orientation. Permafrost zonation and terrain ruggedness from Gruber (2012), glacier polygons from Pfeffer et al. (2014), precipitation from Aphrodite (Yatagai et al., 2012), elevation from SRTM30 (Jarvis et al., 2008).

**Figure 2**

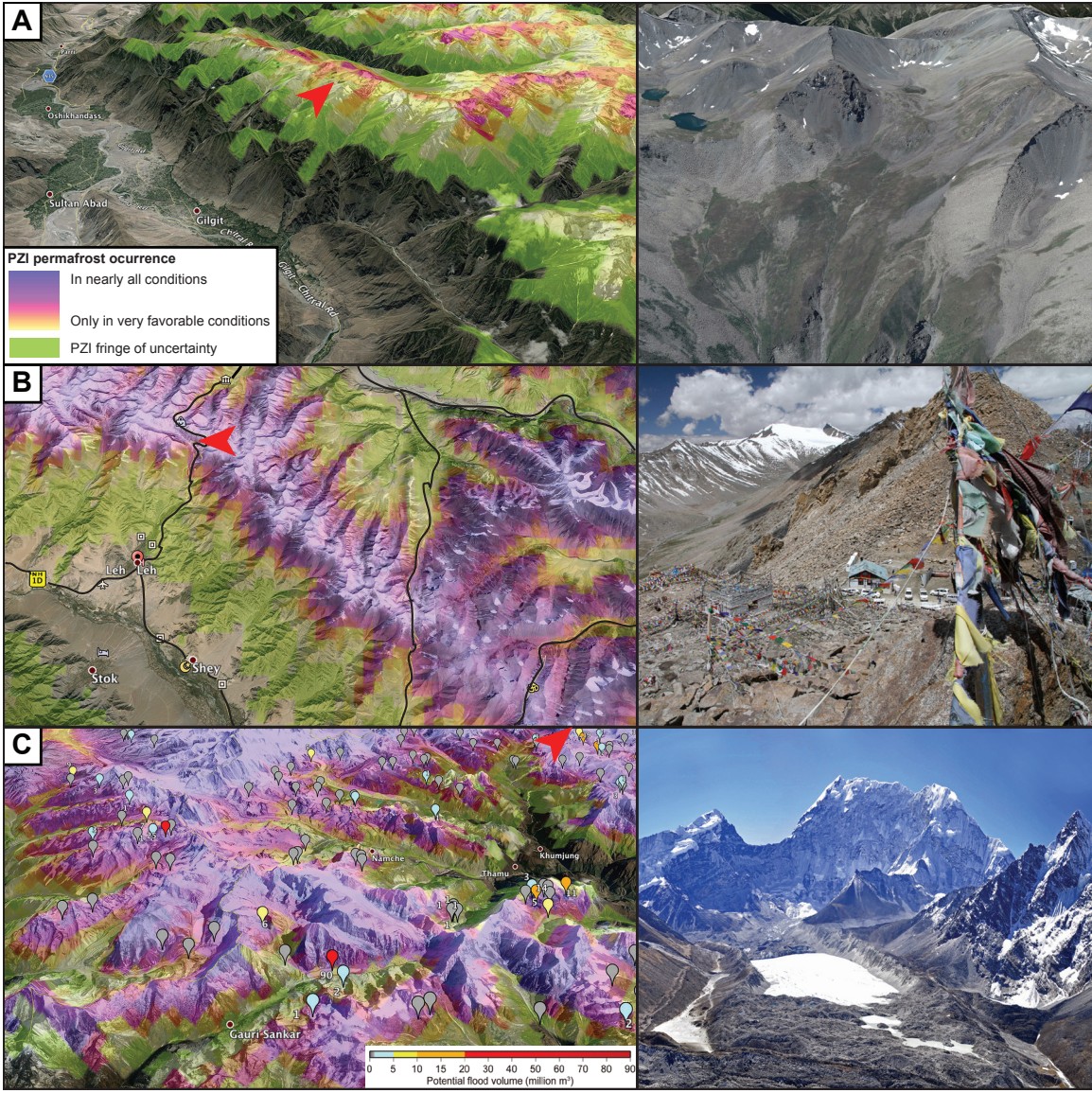

**Figure 2:** Examples of permafrost landscapes in the HKH. (A) Gilgit, Pakistan (35º55`N, 74º18`E, 1500–4700 m a.s.l.). Many of the high-elevation pastures surrounding Gilgit are likely underlain by permafrost and in direct proximity to rock glaciers. Image source: Google Earth. (B) Leh, India (34º9`N, 77º34`E, 3200–5600 m a.s.l.). Several high-elevation roads (black lines) traverse permafrost terrain. Photograph by Adrian Zgraggen. (C) Mountain ranges on the border between Nepal and China (27º50`N, 86º30`E, 3500–8000m a.s.l.). Many of the high-elevation peaks are expected to have large permafrost bodies. Bubble symbols show the locations of glacial lakes as published by Fujita et al. (2013), color scale indicates potential flood volumes. Photograph by Sharad Joshi. All images on the left are derived from Google Earth with overlain Permafrost Zonation Index (PZI, Gruber, 2012). The "PZI fringe of uncertainty" is the lowermost estimate of where permafrost could be found, yellow to blue colors correspond to best estimate. The broad zone between green/yellow and blue colors provides an illustration of the degree of uncertainty inherent in estimating where permafrost occurs and where not. In this overlay, glaciers are not excluded from the PZI signature due to its coarse resolution. Photographs on the right illustrate the general character of the landscapes shown and their locations are indicated with red arrows.

**Figure 3**

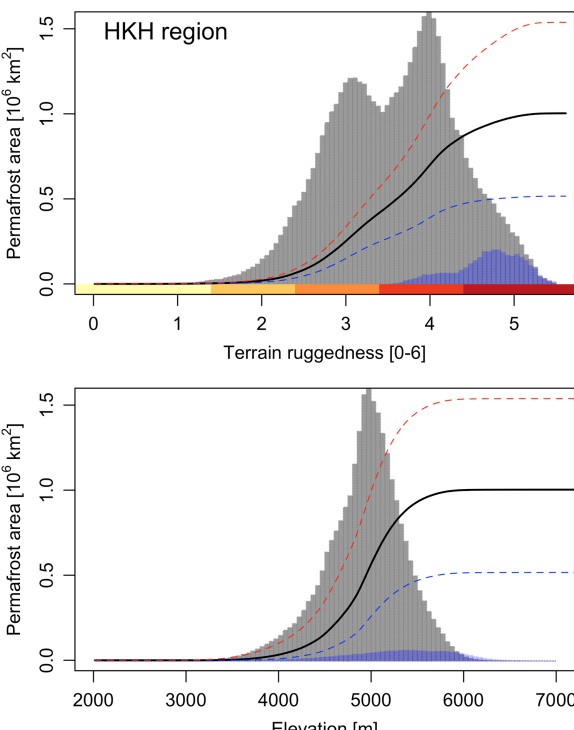

**Figure 3:** Cumulative distribution of estimated permafrost area with respect to terrain elevation and ruggedness (Gruber, 2012). Lines show cumulative permafrost area: best estimate (black), minimum estimate (blue dashed) and maximum estimate (red dashed). Histograms illustrate the distributions of permafrost (grey, best estimate, glacier areas already removed) and glaciers (blue, Pfeffer et al., 2014) but do not correspond to numbers on vertical cumulative axis. Country totals are given in Table 1.

**Figure 4**

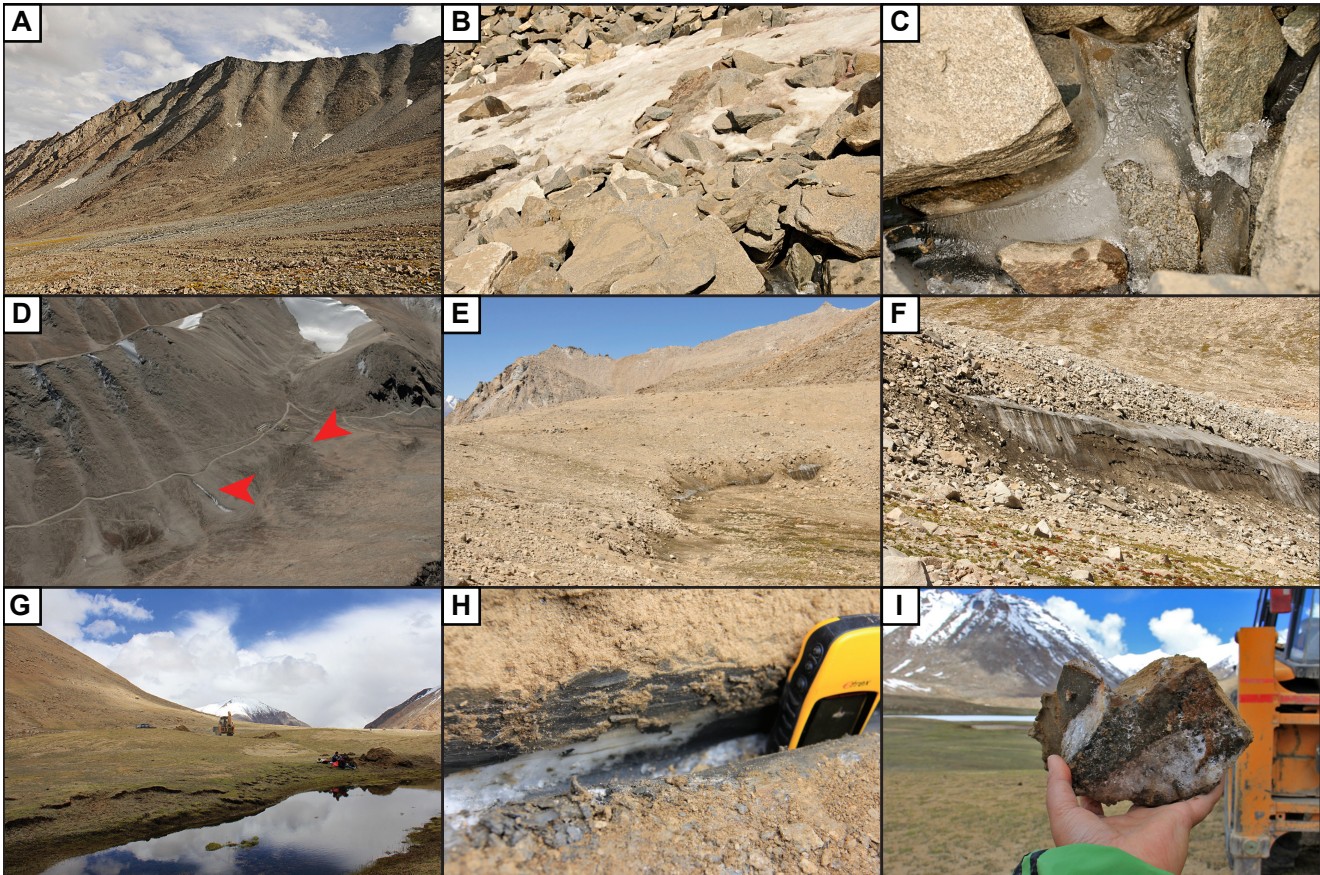

**Figure 4:** Permafrost landscapes and ground ice in Ladakh, India. A: Patches of superimposed ice visible in a north facing slope near Leh, at ~5400 m a.s.l., images B and C provide close-up views. B: Although the melting surface looks like snow, the ice just centimeters below is compact with only few air bubbles. C: Near the ice patches, compact and clear ice with crystal sizes around one cm can be found between blocks close to the surface. D: Chang La pass at ~5400 m a.s.l., red arrows indicate ice patches shown in images E (right) and F (left); the origin and age of the ice patches is unclear. E: Exposure of massive ice near top of pass; note how material above the melting ice cliff appears to slide down, indicating at least partial burial of ice or long-term buttressing exerted by ice patch on adjacent debris. F: Detailed view of ice exposure showing some layering and incorporated clasts. G–I: Excavation in a palaeolake mound near Tsoltak lake, just north of Chang La pass at ~5000 m a.s.l. exposed and sampled a layer of dark and ice-rich sediment at a depth of about 130 cm in June 2015. The sandy soil was ice-cemented from 80 cm downward. Images A–F taken in mid August 2016; image D derived from Google Earth (imagery date: Sep. 26, 2013); images G–I kindly provided by Dr. Binita Phartiyal.