# Peer review of "Review article: Inferring permafrost and permafrost thaw in the mountains of the Hindu Kush Himalaya region"

_The Cryosphere, 2016_

## Short Comment (SC1) · 21 Jun 2016

I read with pleasure the paper of Gruber et al.. I think the paper is generally well written and structured in a clear way.

I propose some minor integration:

Line3, p.3; Studies on recent climate trends from the Himalayan range are limited, and even completely absent at high elevation, At this regard, I suggest to review the recent paper of Salerno et al., 2015 related to the time series of temperature and precipitation reconstructed from seven stations located between 2660 and 5600 m a.s.l. during 1994–2013 on the southern slopes of Mt. Everest.

Line9, p.3. The study of Shrestha et al., 1999 was updated by Kattel and Yao (2013). In western Himalaya (e.g., Bhutiyani et al., 2007; Shekhar et al.,2010), in eastern Himalaya and in the rest of India (e.g., Pal and Al-Tabbaa, 2010). At regional level, on the Tibetan Plateau, I suggest (Liu et al., 2006; Liu et al., 2009). Yang et al. (2006) in the northern part of the central Himalaya. Salerno et al., 2015 in the southern part of the central Himalaya. It is important to point out that winter trends are always found higher in the winter season. In the northern part of Himalaya the warming is observed to be influenced more markedly by the minimum temperature increase (as reported also by Pepin et al., 2015 for the region). In contrast, studies located south of the Himalayan ridge observed a prevalence of maximum increase. In this regard, Salerno et al., 2015 provide a sort of review of these studies. I think that the possible impacts on permafrost (and in general on cryosphere) in HKH are understandable if the scientific community start considering in greater detail the recent finding related to the climatic drivers of change, even if incomplete (considering the remoteness of the region), and even if sometime they do not go in the expected direction (e.g., summer increase of max temp).

Line9, p.3. In relation to the weaking of the monsoon, Salerno at al., 2015 at local scale, but with the unique existing land time series confirm an huge decreasing in the recent years. This analysis is recently confirmed also by Salerno et al., 2016 considering the behavior of glacial unconnected lakes a gridded climatic data.

Line 30, p. 6: In relation to ground temperature measurements in HKH, I suggest to review the findings of Fukui et al., 2007.

Bhutiyani, M. R., Kale, V. S., and Pawar, N. J.: Long-term trends in maximum, minimum and mean annual air temperatures across the Northwestern Himalaya during the twentieth century, Clim. Change, 85, 159–177, doi:10.1007/s10584-006-9196-1, 2007.

Fukui et al., 2007, Changes in the lower limit of mountain permafrost between 1973 and 2004 in the Khumbu Himal, the Nepal Himalayas. Global and Planetary Change

55 (2007) 251–256.

Kattel, D. B. and Yao, T.: Recent temperature trends at mountain stations on the southern slope of the central Himalayas, J. Earth Syst. Sci., 122, 215–227, doi:10.1007/s12040-012-0257-8, 2013.

Liu, K., Cheng, Z., Yan, L., and Yin, Z.: Elevation dependency of recent and future minimum surface air temperature trends in the Tibetan Plateau and its surroundings, Global Planet. Change, 68, 164–174, doi:10.1016/j.gloplacha.2009.03.017, 2009.

Liu, X. D., Yin, Z. Y., Shao, X., and Qin, N.: Temporal trends and variability of daily maximum and minimum, extreme temperature events, and growing season length over the eastern and central Tibetan Plateau during 1961–2003, J. Geophys. Res., 111, 1–19, doi:10.1029/2005JD006915, 2006.

Pal, I. and Al-Tabbaa, A.: Long-term changes and variability of monthly extreme temperatures in India, Theor. Appl. Climatol., 100, 45–56, doi:10.1007/s00704-009-0167-0, 2010.

Salerno, F., Guyennon, N., Thakuri, S., Viviano, G., Romano, E., Vuillermoz, E., Cristofanelli, P., Stocchi, P., Agrillo, G., Ma, Y., and Tartari, G.: Weak precipitation, warm winters and springs impact glaciers of south slopes of Mt. Everest (central Himalaya) in the last 2 decades (1994–2013), Cryosphere, 9, 1229-1247, doi:10.5194/tc-9-1229-2015, 2015.

Salerno, F., Thakuri, S., Guyennon, N., Viviano, G., and Tartari, G.: Glacier melting and precipitation trends detected by surface area changes in Himalayan ponds, The Cryosphere, in press, 2016.

Shekhar, M. S., Chand, H., Kumar, S., Srinivasan, K., and Ganju, A.: Climate-change studies in the western Himalaya, Ann. Glaciol., 51, 105–112, doi:10.3189/172756410791386508, 2010.

Yang, X., Zhang, Y., Zhang, W., Yan, Y., Wang, Z., Ding, M., and Chu, D.: Climate change in Mt. Qomolangma region since 1971, J. Geogr. Sci., 16, 326–336, doi:10.1007/s11442-006-0308-7, 2006.

---

## Referee Comment (RC1) · Anonymous Referee #1 · 7 Jul 2016

The manuscript by Gruber et al. discusses permafrost and permafrost thaw in the mountain of large parts of central Asia. This is a vast region of high elevation, including the Tibet plateau, being the largest permafrost region outside the Arctic low land permafrost. It is clear, that permafrost plays an important role for slope stability, surface water availability, ecology and human activity in that region, and deserves attention.

It is hear the manuscript starts, aiming to discuss general permafrost distribution, potential permafrost thaw following global warming and various impacts following these discussions. However, there are many studies from Chines colleagues especially related to the transport lines over the QTP plateau, and very few permafrost-related studies for the other areas of this vast mountain region. The authors therefore try to transfer

scientific results from other mountain regions, which have more permafrost investigations (mainly or almost exclusively the European Alps) to the central Asian mountains, inferring all sorts of processes and impacts. This arises a general comment:

The manuscript is not a review in a strict sense. It contains passages with text book contents (e.g. p. 3 and pgs. 8/9), and general speculations based on knowledge from other areas. This makes the manuscript interesting reading, e.g. valuable for student courses, and a perfect introduction to a book about Central Asian mountains. I am less convinced about the value in a high-impact scientific journal. It is of course true that we can expect all sorts of impacts if permafrost thaws also in the Hindukush, it is only the question if this statement can be described as "review" or original research.

However, the manuscript of course contains lots of significant information. Important are the real review part, summarizing the work done for the area by the authors or other colleagues. And of course the discussion of the map by Gruber (2012), which is the only higher-resolution permafrost map for the area, providing a good image of the permafrost distribution of the area.

This means, after my opinion, the manuscript is an important contribution, but could be much improved by:

1. Stick to the published investigations, and the map

2. Avoid/reduce substantially the text book passages, explaining basic permafrost/thermal processes etc.

3. Keep the "perspective-part", which was interesting reading. Besides that "promising methodologies" (p. 13, l. 22) maybe also are developed other places then for the Alps. And, if you relate to other mountain areas (which of course is ok within certain limits), maybe other arid mountain ranges as e.g. parts of the Andes etc. could be included more.

4. I would suggest to add some more illustrations, highlighting important work. Now

only derivates from Gruber 2012 are shown more or less.

In summary, I agree that the Hindu Kush Himalaya region is full of "white spots" in terms of understanding permafrost processes there, and that this of course justifies the author's attempt to focus on this region. But I think the manuscript should undergo a thorough revision, focusing more on the "review" part and less on the "inferring" part.

---

## Referee Comment (RC2) · Anonymous Referee #2 · 20 Jul 2016

In this manuscript the authors aim to provide a review of the state of knowledge of permafrost in the Hindu-Kush Himalaya (HKH) region. The paper is motivated by the fact that there is likely a large areal abundance of permafrost in the HKH and therefore widespread impacts are expected under thawing conditions and although research on permafrost in the HKH dates back to the 1970's there are still vast gaps in knowledge. Therefore this review is an important and timely contribution to the topic.

The paper is clear and well written/ structured and does a good job of providing the science basis to the problem as well as referencing existing works in the region. However, I have several comments to put forward for consideration.

MAIN COMMENTS

1. As reviewer#1 already pointed out there are large parts of textbook style text. I would say that the content on P.2 in the chapter "Principles governing...." is useful and important. Physical principles can and should be exchanged between regions while acknowledging that relative importance of processes may change. However, I think that particularly the chapter "Persistence and impacts of permafrost thaw in the HKH" draws too heavily on general permafrost research. The current text is a really nice summary of what can be expected based on known experience/knowledge but could be given more context by rooting in known events/ examples from the HKH in perhaps a more detailed case study style approach. Important topics/events could be GLOFS, landslides/rockfalls, engineering issues. The authors touch on all these topics but in a rather 'high level' manner. These more detailed "case-studies" could also be an opportunity for more figures in the manuscript, which I found quite sparse. I realize the challenge of providing a review over such a diverse area (as you state p7 l1-2) but still feel some more detailed examples would give the manuscript good grounding.

2. I think the chapter on "climate and climate change..." could be expanded as there is certainly a reasonable amount of work in this field in the HKH (as compared permafrost research) and several high elevation initiatives. Climate in the end plays a large part in controlling the distribution and evolution of permafrost and an expanded section would serve as a good starting point for "inferring permafrost". Here it could also be mentioned that the northerly side of the Hindu Kush (Afghanistan) experiences a unimodal precipitation pattern (winter, as shown in Fig1D) dominated by westerlies as effect of monsoon is largely blocked by southern edge of the Hindu-Kush in Pakistan. In this section there could also be an opportunity to discuss snowcover in more detail due to both a good observational record and its important controlling effect on ground temperatures.

3. The climate change part of the the climate chapter could be stronger as there is a good deal of literature on this topic that could be given coverage as this is central to inferring evolution of permafrost in different regions. Example being a discussion of

the diverse effects expected in such a heterogeneous region as the HKH - how could relative changes in precip and temp play out with respect to permafrost in different regions?

4. Although a lot of research questions are presented in the final chapter "Perspectives", I think a useful contribution this paper could make would be to clearly identify and focus on key current knowledge gaps on this topic in the HKH and possible strategies to addressing them. In this context the authors mention simulation and remote sensing (c.f. "Perspectives") but a section on what's needed in terms of ground-based measurements/ networks would be good and how this compliments remote methods by assessing model performance or calibrating RS algorithms.

5. As authors from ICIMOD are present on this paper it might be a good opportunity to see what scope for regional initiatives there are. What's going on currently in this topic (if anything) and what are the possibilities in the future?

TECHNICAL COMMENTS

1. p.3 l.6: "In combination, these effects can cause differences in mean annual ground temperature of more than 10 °C within a distance of less than one kilometer.": reference would be useful here.

2. I think some references from Bodo Bookhagen's precipitation work could be included in the climate section.

3. p.6 l.30 additional measurements references - Ishikawa 2001, Regmi 2008 (you already have these elsewhere).

4. p.9 l21-23 - awkward sentence, perhaps review.

5. p.12 l15-16; "or noticed" sounds odd, perhaps reconsider this sentence.

6. I'm missing Kaab et al 2005 - as an important mountain permafrost hazards reference.

7. Figure 1: Hopefully this can be full page in final publication and perhaps consider adding country outlines (even if these are complex in places) in a subtle way to aid orientation. Glacier outlines are plotted on each subplot but only on legend of 1E I think - I found this slightly confusing at first. Perhaps this legend item should relate to the entire figure.

8. I think figure 2 would be enhanced if you could pairwise match the Google earth/model overlays to existing photos of each example you give, even if its just a small sample of the landscape you show. This would give informative local context.

---

## Author Comment (AC1) · 9 Aug 2016

Reply to comments made by Franco Salerno (doi: 10.5194/tc-2016-104-SC1).

Thank you Dr. Salerno for your effort to comment on our paper in so much detail. Your suggestions have been very interesting and useful for improving our manuscript. We rewrote much of Section "Climate and climate change in the HKH" and inserted more local examples and references (see also the comment of Anonymous Referee 2 and our reply).

Comments by Dr. Salerno are indicated as "SC:", author reply as "AR:". Only sections requiring a reply are reproduced.

SC: [Line 3, page 3] Studies on recent climate trends from the Himalayan range are limited, and even completely absent at high elevation, at this regard, I suggest to review the recent paper of Salerno et al., 2015 related to the time series of temperature and precipitation reconstructed from seven stations located between 2660 and 5600 m a.s.l. during 1994–2013 on the southern slopes of Mt. Everest.

AR: This probably referred to Page 5, Line 3 in the discussion manuscript. We inserted reference to Salerno et al. (2015) here as well as Shea (2015).

SC: [Line 9, page 3] The study of Shrestha et al., 1999 was updated by Kattel and Yao (2013). In western Himalaya (e.g., Bhutiyani et al., 2007; Shekhar et al.,2010), in eastern Himalaya and in the rest of India (e.g., Pal and Al-Tabbaa, 2010). At regional level, on the Tibetan Plateau, I suggest (Liu et al., 2006; Liu et al., 2009). Yang et al. (2006) in the northern part of the central Himalaya. Salerno et al., 2015 in the southern part of the central Himalaya. It is important to point out that winter trends are always found higher in the winter season. In the northern part of Himalaya the warming is observed to be influenced more markedly by the minimum temperature increase (as reported also by Pepin et al., 2015 for the region). In contrast, studies located south of the Himalayan ridge observed a prevalence of maximum increase. In this regard, Salerno et al., 2015 provide a sort of review of these studies. I think that the possible impacts on permafrost (and in general on cryosphere) in HKH are understandable if the scientific community start considering in greater detail the recent finding related to the climatic drivers of change, even if incomplete (considering the remoteness of the region), and even if sometime they do not go in the expected direction (e.g., summer increase of max temp).

AR: We rewrote and expanded most of Section "Climate and climate change in the HKH" taking into account your suggestions and inserted several more local examples and references. The references provided were very useful for this, thank you very much.

SC: [Line 9, page 3] In relation to the weaking of the monsoon, Salerno at al., 2015 at local scale, but with the unique existing land time series confirm an huge decreasing in the recent years. This analysis is recently confirmed also by Salerno et al., 2016 considering the behavior of glacial unconnected lakes a gridded climatic data.

AR: The section on precipitation patterns and Monsoon weakening has been expanded; the differing sign of trends in some simulations is still mentioned.

---

## Author Comment (AC2) · 9 Aug 2016

Reply to comments made by Anonymous Referee #1 (doi:10.5194/tc-2016-104-RC1).

We thank Anonymous Referee #1 for their review and suggestions for improvement. Some of the comments were quite general and we hope our interpretation or implementation are to the point.

Referee comments indicated as "RC:", author reply as "AR:". Only sections requiring a reply are reproduced.

RC: The manuscript is not a review in a strict sense. It contains passages with textbook contents (e.g. p. 3 and pgs. 8/9), and general speculations based on knowledge from

other areas. This makes the manuscript interesting reading, e.g. valuable for student courses, and a perfect introduction to a book about Central Asian mountains. I am less convinced about the value in a high-impact scientific journal. It is of course true that we can expect all sorts of impacts if permafrost thaws also in the Hindukush, it is only the question if this statement can be described as "review" or original research.

AR: We agree that the manuscript is largely not a Summary of permafrost research in the HKH. Because there are not enough local studies available, it reviews a larger body of literature and makes inference on what may be relevant in the HKH. We review existing knowledge and provide a synthesis for a new field of application. This, in our mind, is the essence of a review, which otherwise would be a mere summary. During the access review, the editor has already raised this point and we have subsequently changed the title by introducing "inferring". This should make it more transparent that we are not reviewing work in the HKH but work relevant to the HKH based on current understanding.

RC: However, the manuscript of course contains lots of significant information. Important are the real review part, summarizing the work done for the area by the authors or other colleagues. And of course the discussion of the map by Gruber (2012), which is the only higher-resolution permafrost map for the area, providing a good image of the permafrost distribution of the area. This means, after my opinion, the manuscript is an important contribution, but could be much improved by: 1. Stick to the published investigations, and the map 2. Avoid/reduce substantially the text book passages, explaining basic permafrost/thermal processes etc.

AR: Both points (1 and 2) are best commented together: We believe the value of this text is exactly in the informed interpretation of existing knowledge from outside the HKH to phenomena in the HKH. Some aspects of permafrost science deal with local phenomena, others however, are transferable like other laws in earth science of physics. We believe that (a) the specific composition of these passages is tailored to the HKH and contains a mix of insight derived from both polar and high-elevation permafrost

research, and (b) the argumentation helps to show what special phenomena we can expect in the HKH and what is already know from other locations. We hope this argument will satisfy this suggestion for improvement without changing the text and also point to the response of Anonymous Referee #2 who found this section useful and important.

RC: 3. Keep the "perspective-part", which was interesting reading. Besides that "promising methodologies" (p. 13, l. 22) maybe also are developed other places then for the Alps. And, if you relate to other mountain areas (which of course is ok within certain limits), maybe other arid mountain ranges as e.g. parts of the Andes etc. could be included more.

AR: We have slightly modified the sentence and included two more recent and important non-Alpine references: "Promising methodologies for improved simulation in remote locations and mountain areas exist (Fiddes and Gruber, 2012, 2014; Fiddes et al., 2015; Westermann et al., 2015) and offer synergies with efforts in atmospheric sciences (Gutmann et al., 2016; Ménégoz et al., 2013), glaciology, and hydrology.". In fact, however, most developments in permafrost simulation for mountains did originate form the Alps in the last decade. Concerning other arid mountain ranges, about 15 references to local studies in the Andes, the Rockies, and Central Asia are already used in the text.

RC: 4. I would suggest to add some more illustrations, highlighting important work. Now only derivates from Gruber 2012 are shown more or less.

AR: It is difficult to add useful illustrations or even photographs without going into much case-specific speculation (Is there permafrost or not? Was permafrost really relevant for this vent?) – or showing just rock glaciers (and even there will be debate as to their status as permafrost landforms). Therefore, we have decided to keep the illustrations of this review rather minimal. We have expanded Figure 2 by three photographs (cf. response to Anonymous Referee #2).
RC: In summary, I agree that the Hindu Kush Himalaya region is full of "white spots" in terms of understanding permafrost processes there, and that this of course justifies the author's attempt to focus on this region. But I think the manuscript should undergo a thorough revision, focusing more on the "review" part and less on the "inferring" part.

AR: Thank you, we have done a thorough revision, also in response to the comments made by Anonymous Referee 2 and F. Salerno.

---

## Author Response (AR1)

**Rebuttal letter and revised manuscript**

Dear Editor

This document contains or point-by-point responses to the comments of both Referees, as well as the comment by F. Salerno. These are identical to the Author Comments we have posted in the Interactive Discussion. Referee or short comments are shown in **bold type**, author reply in roman type. Only sections requiring a reply are reproduced.

At the end of this document, the revised manuscript is attached as a PDF with changes highlighted.

The comments by Anonymous Referee #1 were quite vague and it was difficult to distil how we could improve the paper in response to this review. Anonymous Referee #2 has been very helpful and we have made quite a lot of changes as proposed. In some instances, we have argued for a view that differs from that of the referees and we hope that this clarification will make our choices more transparent. We argue this review to be valuable despite (because of) the lack of research in the HKH, and that more case studies will be difficult to include as they would either (a) speculate on permafrost in the absence of research, or (b) be largely biased to the engineering corridor on the QTP and thus miss the main point of the manuscript: the amount and diversity of white spots on the 'permafrost map' in HKH mountains.

Given that several additions were asked in terms of expanding some sections, we hope that exceeding 100 references can be tolerated. It has been difficult to stay within 100 references, as we review a very broad body of permafrost literature, and additionally, climate and environmental background for the HKH.

We thank you and the three referees/commenters for your effort in evaluating and improving this manuscript.

Kind regards
Stephan Gruber, on behalf of all authors

**Reply to comments made by Anonymous Referee #1**

**The manuscript is not a review in a strict sense. It contains passages with textbook contents (e.g. p. 3 and pgs. 8/9), and general speculations based on knowledge from other areas. This makes the manuscript interesting reading, e.g. valuable for student courses, and a perfect introduction to a book about Central Asian mountains. I am less convinced about the value in a high-impact**

**scientific journal. It is of course true that we can expect all sorts of impacts if permafrost thaws also in the Hindukush, it is only the question if this statement can be described as "review" or original research.**

We agree that the manuscript is largely not a *Summary of permafrost research in the HKH*. Because there are not enough local studies available, it reviews a larger body of literature and makes inference on what may be relevant in the HKH. We review existing knowledge and provide a synthesis for a new field of application. This, in our mind, is the essence of a review, which otherwise would be a mere summary. During the access review, the editor has already raised this point and we have subsequently changed the title by introducing "inferring". This should make it more transparent that we are not reviewing work in the HKH but work relevant to the HKH based on current understanding.

**However, the manuscript of course contains lots of significant information. Important are the real review part, summarizing the work done for the area by the authors or other colleagues. And of course the discussion of the map by Gruber (2012), which is the only higher-resolution permafrost map for the area, providing a good image of the permafrost distribution of the area. This means, after my opinion, the manuscript is an important contribution, but could be much improved by:**
**1. Stick to the published investigations, and the map**
**2. Avoid/reduce substantially the text book passages, explaining basic permafrost/thermal processes etc.**
Both points (1 and 2) are best commented together: We believe the value of this text is exactly in the informed interpretation of existing knowledge from outside the HKH to phenomena in the HKH.  Some aspects of permafrost science deal with local phenomena, others however, are transferable like other laws in earth science of physics. We believe that (a) the specific composition of these passages is tailored to the HKH and contains a mix of insight derived from both polar and high-elevation permafrost research, and (b) the argumentation helps to show what special phenomena we can expect in the HKH and what is already know from other locations.  We hope this argument will satisfy this suggestion for improvement without changing the text and also point to the response of Anonymous Referee #2 who found this section useful and important.

**3. Keep the "perspective-part", which was interesting reading. Besides that "promising methodologies" (p. 13, l. 22) maybe also are developed other places then for the Alps. And, if you relate to other mountain areas (which of course is ok within certain limits), maybe other arid mountain ranges as e.g. parts of the Andes etc. could be included more.**
We have slightly modified the sentence and included two more recent and important non-Alpine references: "Promising methodologies for improved simulation in remote locations and mountain areas exist (Fiddes and Gruber, 2012, 2014; Fiddes et al., 2015; Westermann et al., 2015) and offer synergies with efforts in atmospheric sciences (Gutmann et al., 2016; Ménégoz et al., 2013), glaciology, and hydrology.". In fact, however, most developments in permafrost simulation for

mountains did originate form the Alps in the last decade. Concerning other arid mountain ranges, about 15 references to local studies in the Andes, the Rockies, and Central Asia are already used in the text.

**4. I would suggest to add some more illustrations, highlighting important work. Now only derivates from Gruber 2012 are shown more or less.**
It is difficult to add useful illustrations or even photographs without going into much case-specific speculation (Is there permafrost or not? Was permafrost really relevant for this vent?) – or showing just rock glaciers (and even there will be debate as to their status as permafrost landforms). Therefore, we have decided to keep the illustrations of this review rather minimal.  We have expanded Figure 2 by three photographs (cf. response to Anonymous Referee #2).

**In summary, I agree that the Hindu Kush Himalaya region is full of "white spots" in terms of understanding permafrost processes there, and that this of course justifies the author's attempt to focus on this region. But I think the manuscript should undergo a thorough revision, focusing more on the "review" part and less on the "inferring" part.**
Thank you, we have done a thorough revision, also in response to the comments made by Anonymous Referee 2 and F. Salerno.

**Reply to comments made by Anonymous Referee #2**

**MAIN COMMENT 1. As reviewer#1 already pointed out there are large parts of textbook style text. I would say that the content on P.2 in the chapter "Principles governing...." is useful and important. Physical principles can and should be exchanged between regions while acknowledging that relative importance of processes may change. However, I think that particularly the chapter "Persistence and impacts of permafrost thaw in the HKH" draws too heavily on general permafrost research. The current text is a really nice summary of what can be expected based on known experience/knowledge but could be given more context by rooting in known events/ examples from the HKH in perhaps a more detailed case study style approach. Important topics/events could be GLOFS, landslides/rockfalls, engineering issues. The authors touch on all these topics but in a rather 'high level' manner. These more detailed "case-studies" could also be an opportunity for more figures in the manuscript, which I found quite sparse. I realize the challenge of providing a review over such a diverse area (as you state p7 l1-2) but still feel some more detailed examples would give the manuscript good grounding.**
Thank you for acknowledging the value of passages with textbook character in this manuscript. We agree that it would be beneficial to scaffold and illustrate Section "Persistence and impacts of permafrost thaw in the HKH" with case studies and images and have invested a lot of thought into this. At the same time, we would be speculating on the existence and role of permafrost in the absence of published

studies (cf. reply to Anonymous Referee #2 with respect to inserting more images). Figure 2 is the best we thought we could do while making the uncertain character of permafrost estimates tangible via the model colors and the visible course-resolution grid: Part A refers to pastures (without having published evidence of the existence or role of permafrost at that location), Part B refers to engineering issues (roads in permafrost terrain) without having published evidence, Part C refers to the additional risk of GLOFs presented by large frozen summits over lakes as well as ground ice in dams without having to make a detailed argument at one location. We have now included photos in the new version of this figure.

Below is another figure we produced but then decided not to use. This was because it selects two types of landscapes and thus possibly shapes the perception of impacts more restrictedly than what is useful. With this, we hope to illustrate here that we are not lightly brushing this comment aside, but rather have chosen the current very lean form of illustration in the manuscript with great care.

[Figure]

Figure 1: Possible impacts of permafrost exemplified in sample catchments: **(A)** Impacts on hydrology and water quality. 1) Drying near-surface soil caused by deepening active layer. 2) Drying of depressions and small creeks due to increasing deep percolation in catchment. 3) Increased soil moisture and discharge in small creeks due to meltwater release from ice-rich ground. 4) Increased solute availability through release from ground ice melt and leaching of newly permeable soil. 5) More base flow in larger streams and additional discharge from melt of old ground ice during and after hot summers. 6) Changed water chemistry in streams draining catchments with large permafrost extent. Vegetation will respond to changes in moisture and nutrient availability. This will likely results in large-scale reduction in vegetation density overprinted with local and transient greening. **(B)** Impacts on geohazards. 7) Large rock/ice fall or landslides with far reach. 8) Small rockfall. 9) Debris flows from thawing debris or remobilized rock fall deposits. 10) Lake outburst flood following impact of rock/ice fall. 11) Outburst flood following ground ice loss in lake dam. 12) Changes in stream geometry following increased

sediment load. The environments and configurations for actual impacts may differ from the examples shown. Other impacts described in text. Images derived from Google Earth.

**MAIN COMMENT 2. I think the chapter on "climate and climate change..." could be expanded as there is certainly a reasonable amount of work in this field in the HKH (as compared permafrost research) and several high elevation initiatives. Climate in the end plays a large part in controlling the distribution and evolution of permafrost and an expanded section would serve as a good starting point for "inferring permafrost". Here it could also be mentioned that the northerly side of the Hindu Kush (Afghanistan) experiences a unimodal precipitation pattern (winter, as shown in Fig1D) dominated by westerlies as effect of monsoon is largely blocked by southern edge of the Hindu-Kush in Pakistan. In this section there could also be an opportunity to discuss snowcover in more detail due to both a good observational record and its important controlling effect on ground temperatures.**
The chapter on climate and climate change has been significantly expanded based on this comment and the comments of F. Salerno. We also added: "In the far west of the HKH, a unimodal pattern of cyclonic winter precipitation can be found on the northern side of the Hindu Kush (Schiemann et al., 2008)." A new paragraph on snow cover has been inserted into Section "Climate and climate change in the HKH".

**MAIN COMMENT 3. The climate change part of the climate chapter could be stronger as there is a good deal of literature on this topic that could be given coverage as this is central to inferring evolution of permafrost in different regions. Example being a discussion of the diverse effects expected in such a heterogeneous region as the HKH - how could relative changes in precip and temp play out with respect to permafrost in different regions?**
This section has been strongly expanded and this includes more information on observed climate change and its patterns (regional, elevation, season). As for the consequences in different regions, the mechanisms are explained in the text. Making regional statements is difficult given the fact that differing landscape facets in one region already may show very different reactions. For this reason, a more general statement along these lines was inserted into Section "Persistence and impacts of permafrost thaw in the HKH": "Climate change (air temperature, precipitation, cloudiness...) in the HKH exhibits diverse regional patterns as well as differing trends at high/low elevation or during differing seasons. As these changes are further overprinted by topography, the resulting effect on permafrost is likely to be spatially highly diverse as well."

**MAIN COMMENT 4. Although a lot of research questions are presented in the final chapter "Perspectives", I think a useful contribution this paper could make would be to clearly identify and focus on key current knowledge gaps on this topic in the HKH and possible strategies to addressing them. In this context the authors mention simulation and remote sensing (c.f. "Perspectives") but a section on what's needed in terms of ground-based**

**measurements/ networks would be good and how this compliments remote methods by assessing model performance or calibrating RS algorithms.**
The clear identification of key knowledge gaps requires prioritizing some issues over others and thus attaching value to differing outcomes (e.g., scientific versus practical relevance or large engineering projects versus traditional livelihoods). Many of the rather diverse research questions presented have been developed by the authors and many others have been "…inspired by the outcomes of a group discussion during the International Symposium on Glaciology in High Mountain Asia, Kathmandu, Nepal, in March 2015." (Acknowledgements). We feel that a prioritization may be too subjective and speculative to be useful here.
We agree that a high-level description of ground-based measurements and networks would be very useful and, at the same time, we do not want to be too descriptive on its form or implementation We have changed a sentence in the beginning of the Perspective Section: "Long-term monitoring of ground temperature, ice content, and other variables at selected sites (cf., Vonder Mühll et al., 2008) will contribute to national and international programs and provide a basis for the evaluation of simulation and remote-sensing products." and included a reference to similar activity.

**MAIN COMMENT 5. As authors from ICIMOD are present on this paper it might be a good opportunity to see what scope for regional initiatives there are. What's going on currently in this topic (if anything) and what are the possibilities in the future?**
In our opinion, this would challenge the scope of a scientific article. For the context of this reply, ICIMOD has been exploring and preparing regional initiatives via its Permafrost pilot study (http://www.icimod.org/?q=11478) and held a corresponding Scoping Meeting in 2015 (http://www.icimod.org/?q=13932). Some initiatives are starting and interested parties should contact:
    Dorothea Stumm
    Senior Glaciologist and Permafrost Project Coordinator
    dorothea.stumm@icimod.org

**TECHNICAL COMMENT 1. p.3 l.6: "In combination, these effects can cause differences in mean annual ground temperature of more than 10 ̲C within a distance of less than one kilometer.": reference would be useful here.**
The statement can now be better traced based on three references given "(Gruber et al., 2004b; Gubler et al., 2011; Hasler et al., 2011b)".

**TECHNICAL COMMENT 2. I think some references from Bodo Bookhagen's precipitation work could be included in the climate section.**
Included Bookhagen and Burbank (2006, 2010).

**TECHNICAL COMMENT 3. p.6 l.30 additional measurements references - Ishikawa 2001, Regmi 2008 (you already have these elsewhere).**

The sentence specifically refers to ground temperature measurements. Ishikawa at al. (2001) report rock glacier distribution, air temperature and geophysics; Regmi (2008) reports rock glacier distribution.

**TECHNICAL COMMENT 4. p.9 l21-23 - awkward sentence, perhaps review.**
Restructured and shortened into: "In summary, permafrost thaw results in an increased frequency and possibly in unexpected locations of mass movements, as well as in increased sediment loads available for further downstream transport."

**TECHNICAL COMMENT 5. p.12 l15-16; "or noticed" sounds odd, perhaps reconsider this sentence.**

**TECHNICAL COMMENT 6. I'm missing Kaab et al 2005 - as an important mountain permafrost hazards reference.**
Yes, indeed. The hazard paragraph of section 'Persistence and impacts of permafrost thaw in the HKH' now finishes with: "Often, processes related to glaciers and to permafrost conspire in producing high-mountain hazards. These represent a continuous threat to human lives and infrastructure and related disasters can kill thousands of people at once and cause damage on the order of 100 million dollars per year, globally (Kääb et al., 2005)."

**TECHNICAL COMMENT 7. Figure 1: Hopefully this can be full page in final publication and perhaps consider adding country outlines (even if these are complex in places) in a subtle way to aid orientation. Glacier outlines are plotted on each subplot but only on legend of 1E I think - I found this slightly confusing at first. Perhaps this legend item should relate to the entire figure.**
Thank you, we agree on the glacier outlines with respect to the legend. We have changed the figure and hope it is now less confusing. Concerning country outlines: The figures already contain very much information, and including boundaries would make them even denser. We have considered adding them but decided against it, thereby also following ICIMOD's publication policy for this rather delicate issue. Names of countries are given in 1A, which we hope will be sufficient.

**TECHNICAL COMMENT 8. I think figure 2 would be enhanced if you could pairwise match the Google earth/model overlays to existing photos of each example you give, even if its just a small sample of the landscape you show. This would give informative local context.**
Done.

**Replies to comments made by Franco Salerno**

**[Line 3, page 3] Studies on recent climate trends from the Himalayan range are limited, and even completely absent at high elevation, at this regard, I**

**suggest to review the recent paper of Salerno et al., 2015 related to the time series of temperature and precipitation reconstructed from seven stations located between 2660 and 5600 m a.s.l. during 1994–2013 on the southern slopes of Mt. Everest.**

This probably referred to Page 5, Line 3 in the discussion manuscript. We inserted reference to Salerno et al. (2015) here as well as Shea (2015).

**[Line 9, page 3] The study of Shrestha et al., 1999 was updated by Kattel and Yao (2013). In western Himalaya (e.g., Bhutiyani et al., 2007; Shekhar et al.,2010), in eastern Himalaya and in the rest of India (e.g., Pal and Al-Tabbaa, 2010). At regional level, on the Tibetan Plateau, I suggest (Liu et al., 2006; Liu et al., 2009). Yang et al. (2006) in the northern part of the central Himalaya. Salerno et al., 2015 in the southern part of the central Himalaya. It is important to point out that winter trends are always found higher in the winter season. In the northern part of Himalaya the warming is observed to be influenced more markedly by the minimum temperature increase (as reported also by Pepin et al., 2015 for the region). In contrast, studies located south of the Himalayan ridge observed a prevalence of maximum increase. In this regard, Salerno et al., 2015 provide a sort of review of these studies. I think that the possible impacts on permafrost (and in general on cryosphere) in HKH are understandable if the scientific community start considering in greater detail the recent finding related to the climatic drivers of change, even if incomplete (considering the remoteness of the region), and even if sometime they do not go in the expected direction (e.g., summer increase of max temp).**

We rewrote and expanded most of Section "Climate and climate change in the HKH" taking into account your suggestions and inserted several more local examples and references. The references provided were very useful for this, thank you very much.

**[Line 9, page 3] In relation to the weaking of the monsoon, Salerno at al., 2015 at local scale, but with the unique existing land time series confirm an huge decreasing in the recent years. This analysis is recently confirmed also by Salerno et al., 2016 considering the behavior of glacial unconnected lakes a gridded climatic data.**

The section on precipitation patterns and Monsoon weakening has been expanded; the differing sign of trends in some simulations is still mentioned.

**Revised manuscript with changes highlighted attached below**

[revised manuscript text omitted]

Stephan Gruber 2016-7-23 11:16

Stephan Gruber 2016-7-23 11:08

Stephan Gruber 2016-7-23 20:37

Stephan Gruber 2016-7-23 19:22

Stephan Gruber 2016-7-23 10:03

Stephan Gruber 2016-7-23 21:36

Stephan Gruber 2016-7-29 20:50

Stephan Gruber 2016-7-23 21:35
**Moved (insertion) [1]**

Stephan Gruber 2016-7-23 21:35
**Moved up [1]:** (Shrestha et al., 1999; Yang et al., 2011)

Stephan Gruber 2016-7-23 21:36

Stephan Gruber 2016-7-23 21:25

Stephan Gruber 2016-7-29 20:40

Stephan Gruber 2016-7-29 20:45

Stephan Gruber 2016-7-30 09:31
**Moved (insertion) [2]**

Stephan Gruber 2016-8-8 13:09

Stephan Gruber 2016-8-8 13:09

Stephan Gruber 2016-7-30 09:31

[revised manuscript text omitted]

---

## Author Response (AR2)

Dear Nadine,

Thank you again for your comments and suggestions to improve this manuscript. Our replies are included below and the revised manuscript with tracked changes is attached.

Furthermore, as a number of images became available this summer, we have included some of them, addressing a reviewer request that earlier we could not address as well as we can now.

Kind regards,
Stephan

**Editor Comment 1)** The suggestion by F. Salerno and Reviewer 2, to include more climate and climate related studies has been followed well by the authors in the revised manuscript. However, an expert assessment which implications these changes might have for the permafrost evolution in the HKH is not provided. Or to express it in the words of Reviewer 2 'the chapter of climate and climate change could be expanded …. and serve as a good starting point for 'inferring permafrost' ''. It is fully clear, that such an 'inferring' will be associate with some speculations, which the authors in principle want to avoid, as written in your replies. However at the same time, I believe 'inferring' in the sense it is used for this article, automatically includes also some 'speculation'?! In any case, it seems to be a clear expectation by the readers to get some thoughts from the experts (that are the authors of this article) in this regard. Moreover, and probably most importantly, with some physically plausible and carefully formulated speculations (which should be clearly marked as such) by the authors, awareness rising for the topic of permafrost by the local researchers and authorities would be much higher. A suggestion how to do this, is given by Ref 2; by 'case-study based climate change impact assessments'. I am sure this is also in the sense of Reviewer 1, who recommended reducing the 'text book passages' and 'stick to published investigation, and the map'.

**Reply:** The manuscript already contains some speculation on the present state (*"Permafrost likely follows known patterns in many ways: given the same MAAT, a larger abundance of permafrost is expected in arid areas than in areas with more precipitation. In humid areas, few debris slopes are expected to have permafrost with the exception of windblown areas devoid of snow, coarse block covers, peat lands, or debris with incorporated meteoric ice. In more arid areas, extensive permafrost occurrence is to be expected, even below vegetated surfaces. The distribution of permafrost in steep bedrock is much less affected by precipitation as sliding controls snow amounts. The effect of monsoonal precipitation regimes with wet summers and dry winters (Figure 1D) on the differentiation of ground temperature with respect to MAAT is unknown. Possible effects can include reversed thermal offset (Lin et al., 2015) as well as a changed net effect of snow cover, providing little insulation in winter and*

*frequent increases in albedo and latent heat in summer. [...] As a consequence, the temperature difference between shaded and sun-exposed slopes at extreme elevations likely exceeds known values. [...] The burial of snow or avalanche deposits (Gruber and Haeberli, 2009) in talus slopes and cones is expected to cause large aggrading bodies of permafrost. [...] Buried ice can occur near the margins of present-day glaciers and in formerly glacierized areas. When this occurs in cold permafrost, buried ice can persist for long times and thus allow surface processes to gradually obliterate surface expressions of buried ice."*). This has now been expanded by "*Strong winter cooling can furthermore support local aggradation by surface melt water refreezing in soil macro-pores or in snow accumulations generated by wind drifting or avalanches (superimposed ice). Corresponding bodies of ice are visible on the surface and in the shallow subsurface (Figure 4A–F) and can be expected to exist more frequently at depth.*" and the new Figure 4.

In your Editor Comment you write "*…implications these changes might have…"*, likely pointing to the implications of climate change on permafrost. We have now added more text on this: "*Some speculation on the effect of the diverse regional changes can inform future research: Winter warming, for instance, may have a weaker effect on ground temperatures than warming during the summer months due to the insulation by snow. Short-lived climate pollutants such as black carbon, dust, and aerosols can shorten the duration of the snow cover (Ménégoz et al., 2014) and thus exert a warming effect on the ground by exposing it earlier to warming surface fluxes. A delayed onset of the snow cover in areas subject to winter westerlies may promote increased ground cooling.*". But honestly, this already stretches what I feel comfortable with speculating. I have deliberately not included a speculation on a possible weakening of high-elevation warming by reduced latent heat release and long-wave emission in a weakening Monsoon. This is because I do not want to venture too far outside my core area of expertise.

I hope you understand that a 'case-study based climate change impact assessments' (Reviewer 2) really is beyond the scope of our manuscript. Similarly, if we 'stick to published investigation, and the map', (Reviewer 1) we would not speculate at all. I hope the changes we made strike a good balance and address your concerns more fully than in our last revision.

**Editor Comment 2)** A second point which in my view has not been adequately considered, but is clearly expected by the readers (and again helps to raise awareness…) is the point raised by Ref 2., that ongoing initiatives should be mentioned. For instance, some sentences could be included at the end of the ^Perspective' chapter. As raised by Ref 2, ICIMOD (which is co-authoring this article) has several initiatives ongoing, and also in other programmes, initial initiatives in terms of permafrost have been started and just recently been communicated (Allen et al. 2016). The authors mention that the data that has been collected so far is not able to show anything related to PF, as these are only short term GST records and that it would be premature to show these data. I agree with this, but these activities can/should still be mentioned, also without showing any data records from these ongoing investigations.

**AC:** Given that our previous argument not to include this did not alleviate your concern, we have now included a new paragraph into section 'Perspectives' as you have suggested.

[revised manuscript text omitted]

Stephan Gruber 2016-10-18 06:56

Stephan Gruber 2016-10-18 06:56